# Contributions of residential coal combustion to the air quality in Beijing-Tianjin-Hebei (BTH), China: A case study

Xia Li[1,2], Jiarui Wu[1], Miriam Elser[3], Tian Feng[1], Junji Cao[1], Imad El-Haddad[3], Rujin Huang[1], Xuexi Tie[3], André S. H. Prévôt[3], and Guohui Li[1*]

[1]Key Lab of Aerosol Chemistry and Physics, SKLLQG, Institute of Earth Environment, Chinese Academy of Sciences, Xi'an, China
[2]University of Chinese Academy of Science, Beijing, China
[3]Laboratory of Atmospheric Chemistry, Paul Scherrer Institute, 5232 Villigen, Switzerland

[*]Correspondence to: Guohui Li (ligh@ieecas.cn)

**Abstract:** In the present study, the WRF-Chem model is used to assess contributions of residential coal combustion (RCC) emission to the air quality in Beijing-Tianjin-Hebei (BTH) during a persistent air pollution episode from 9 to 25 January 2014. In general, the predicted temporal variations and spatial distributions of the mass concentrations of air pollutants are in good agreements with observations at monitoring sites in BTH. The WRF-Chem model also reasonably reproduces the temporal variations of aerosol species when compared with the AMS measurements in Beijing. The RCC emission plays an important role in the haze formation in BTH, contributing about 23.1% of $PM_{2.5}$ (fine particulate matter) and 42.6% of $SO_2$ during the simulation period on average. Organic aerosols dominate the $PM_{2.5}$ from the RCC emission in BTH, with a contribution of 42.8%, followed by sulfate (17.1%). The air quality in Beijing is remarkably improved when the RCC emission in BTH and its surrounding areas is excluded in model simulations, with a 30% decrease of $PM_{2.5}$ mass concentrations. However, if only the RCC emission in Beijing is excluded, the local $PM_{2.5}$ mass concentration is decreased by 18.0% on average. Our results suggest that the implementation of the residential coal replacement by clean energy sources in Beijing is beneficial to the local air quality. Should the residential coal replacement be carried out in BTH and its surrounding areas, the air quality in Beijing would be improved remarkably. Further studies would need to consider uncertainties in the emission inventory and meteorological fields.

## 1    Introduction

Over the several past decades, China has experienced rapid economic growth, accompanied with accelerating industrialization and urbanization, which has seriously deteriorated air quality (e.g., Zhang et al., 2009; Zhang et al., 2012; Zhang et al., 2015). Recently, haze pollution has become the primary concern about air quality in most key regions and cities in China, especially in Beijing-Tianjin-Hebei (BTH) and Yangtze River Delta (YRD) (e.g., Wang et al., 2005; An et al., 2007; Wang et al., 2014; Chen et al., 2016; Gao et al., 2016). The severe and persistent haze pollution with high concentrations of fine particulate matter ($PM_{2.5}$) and the consequent low visibility, is mainly caused by heavy anthropogenic emissions and unfavorable synoptic situations (e.g., Seinfeld and Pandis, 2006; Lei et al., 2011; Lv et al., 2016; Wang et al., 2016; Ziková et al., 2016). According to the China's Ministry of Environment Protection (MEP), the annual mean mass concentration of $PM_{2.5}$ was 102 μg m$^{-3}$ in 2013 and 93 μg m$^{-3}$ in 2014 in BTH, far beyond the World Health Organization (WHO) interim target-1 of 35 μg m$^{-3}$ for the annual mean $PM_{2.5}$ mass concentration and the secondary class standard in China's new National Ambient Air Quality Standard (NAAQS, GB3095-2012). Therefore, in order to improve the air quality in BTH, the Chinese State Council has issued the "Atmospheric Pollution Prevention and Control Action Plan" (APPCAP) in September 2013 to reduce $PM_{2.5}$ by 25% by 2017 relative to 2012 levels. Since implementation of the APPCAP, stringent control strategies have been carried out to reduce pollutants emissions from power plants, industries and transportation (Sheehan et al., 2014; Liu et al., 2015; Yang et al., 2016). Control strategies have also been implemented to reduce residential emissions, but evaluation means constrained by observations are still lacking.

The air pollution in China is a typical coal-smoke pollution, which is considered to be closely associated with China's special energy consumption structure (e.g., Quan et al., 2014; Archernicholls et al., 2016; Liu et al., 2016; Xue et al., 2016). Coal plays a key role in China's

energy structure, and as the most abundant and a relatively cheap energy resource, coal is regarded as a dominant energy supply in China in the foreseeable future. According to the BP statistical review of world energy, from 1980s to the present day, the proportion of coal in China' primary energy production and consumption has been around 70%, which is much higher than that of around 20% in OECD (Organization for Economic Co-operation and Development) countries. Entering the 21 centuries, coal consumption in China has increased sharply, and by 2013, China's coal consumption has accounted for 50.3% of the global coal consumption, which was 4.2 and 6.7 times higher than that of the United States and European Union, respectively. It is reported that in 2013, coal is responsible for 79%, 54%, 40%, 35%, 40%, and 17% of the $SO_2$, $NO_x$, $PM_{10}$, $PM_{2.5}$, BC, and OC emissions in China, respectively (Ma et al., 2016).

Residential coal combustion (RCC) emission is recognized as a significant source of air pollution, affecting both local and regional air quality and posing serious threat to human health and environment by releasing hazardous air pollutants, including particulate matter (PM), black carbon (BC), organic carbon (OC), $SO_2$, nitrogen oxide ($NO_x$), CO, $CO_2$, and polycyclic aromatic hydrocarbons (e.g., Wornat et al., 2001; Ge et al., 2004; Zhi et al., 2008; Shen et al., 2010; Cheng et al., 2016; Li et al., 2016). Recently, chemical transport models have been used to investigate the contribution of RCC emissions to the ambient air pollution in China. Using the CMAQ model, Xue et al. (2016) have shown that during the winter heating season of 2012, the contribution of RCC emissions in Beijing to the mass concentrations of local $PM_{10}$, $SO_2$, $NO_x$, and CO is 11.6%, 27.5%, 2.8%, and 7.3%, respectively. Simulations using the GEOS-Chem model by Ma et al. (2016) have demonstrated that coal combustion contributes 40% of the total $PM_{2.5}$ mass concentrations on national average in 2013. Among major coal-burning sectors, industrial coal burning contributes 17% of the $PM_{2.5}$ concentrations, followed by power plants (9.8%) and domestic sector (4.0%). Liu et al. (2016) have used the Weather Research

and Forecasting model coupled with chemistry (WRF-Chem) to simulate the air pollution in
BTH in January and February 2010, indicating that annual elimination of residential sources in
BTH reduces emissions of primary $PM_{2.5}$ by 32%, compared with 5%, 6%, and 58% of
transportation, power plants, and industrial sectors, respectively. Using the source-oriented
CMAQ model, Qiao et al. (2017) have conducted simulations to evaluate source apportionment
of $PM_{2.5}$ in 25 Chinese provincial capitals and municipalities and concluded that industrial and
residential sources are predicted to be the largest contributor to $PM_{2.5}$ for all the city groups,
with annual fractional contributions of 25.0%-38.6% and 9.6%-27%, respectively.
Until now, there have been few studies focusing specially on the impacts of RCC
emissions on the air quality in BTH. In the present study, we use the WRF-Chem model to
assess the contribution of RCC emissions to the air quality in BTH during a persistent air
pollution episode from 9 to 25 January 2014. The WRF-Chem model configurations and
methodology are described in Section 2. Model results and discussions are represented in
Section 3, and conclusions are given in Section 4.

**2    Model and Methodology**
**2.1    WRF-Chem model and configurations**
The WRF-Chem model used in this study is developed by Li et al. (2010, 2011a, b, 2012)
at the Molina Center of Energy and Environment, based on previous studies (Grell et al., 2005;
Fast et al., 2006). The wet deposition of aerosols follows the method used in the CMAQ module
and the dry deposition of chemical species is parameterized following Wesely (1989). The
photolysis rates are calculated using the FTUV (fast radiation transfer model), including the
aerosol and cloud effects on photolysis (Tie et al., 2003; Li et al., 2005, 2011a). The inorganic
aerosols are calculated using ISORROPIA Version 1.7 (Nenes, 1998). The secondary organic
aerosol (SOA) is predicted using the volatility basis-set (VBS) modeling method, with
contributions from glyoxal and methylglyoxal.
The WRF-Chem model adopts one grid with a horizontal resolution of 6 km centered at
39°N, 117°E, and 35 sigma vertical levels with a stretched vertical grid with spacing ranging
from 30 m near the surface, to 500 m at 2.5 km and 1 km above 14 km, and the grid cells used
for the domain are 150 × 150. The physical parameterizations employed in the simulation
include the microphysics scheme of Hong and Lim (2006), the unified Noah Land-surface
model (Chen and Dudhia, 2001), the Goddard longwave scheme (Chou and Suarez, 2001), and
the Goddard shortwave scheme (Chou and Suarez, 1999). The National Centers for
Environmental Prediction (NCEP) 1°×1° reanalysis data are used for the meteorological initial
and boundary conditions, and the meteorological simulations are not nudged in the study. The
chemical initial and boundary conditions are interpolated from the 6 h output of MOZART
(Horowitz et al., 2003). The spin-up time of the WRF-Chem model is 28 h. The monthly
average anthropogenic emissions with a 6 km horizontal resolution in the North China Plain
are developed by Zhang et al. (2009) with the base year of 2013, including contributions from
agriculture, industry, power generation, residential, and transportation sources, and the volatile
organic compounds (VOCs) speciation based on the SAPRC99 chemical mechanism. The
temporal allocation for different sources follows those in Zhang et al. (2009). The biogenic
emissions are calculated online using the MEGAN (Model of Emissions of Gases and Aerosol
from Nature) model developed by Guenther et al. (2006).
A persistent air pollution episode from 9 to 25 in January 2014 in BTH is simulated using
the WRF-Chem model. During the study period, the average $PM_{2.5}$ mass concentration in BTH
is 161.9 μg m$^{-3}$, with a maximum of 323.5 μg m$^{-3}$. The average temperature and relative
humidity in Beijing during the period is −1.7 ℃ and 32.3%, respectively, and the average wind
speed is about 2.8 m s$^{-1}$. The model simulation domain is shown in Figure 1, and detailed model
configurations can be found in Table 1.
The brute force method is used to quantify the contribution of the RCC emission in BTH
and its surrounding areas to the air quality (Dunker et al., 1996). It is worth noting that, although
the method can evaluate the importance of the certain emission source, it still has flaws in
quantifying the source contribution, considering the complicated non-linear processes in the
atmosphere (Zhang and Ying, 2011). In the present study, we have conducted one reference
simulation in which emissions from various anthropogenic and biogenic sources are considered
(hereafter referred as to the REF case). The results from the REF case are compared with
observations in BTH to validate the model performance. Additional two sensitivity simulations
have also been performed, without the RCC emission in BTH and its surrounding areas and
Beijing, respectively (hereafter referred as to the SEN-BTH case and SEN-PEK case). In the
sensitivity simulation, the emission of $NO_x$, CO, VOCs, $SO_2$, black and organic carbon,
primary sulfate, and unspecified particulate matters from the RCC is turned off. The difference
between the reference and sensitivity simulation is used to evaluate contributions of RCC
emissions to the air quality in BTH and Beijing.
**2.2    Statistical methods for comparisons**
In the present study, we use the mean bias (MB), root mean square error (RMSE) and
index of agreement (IOA) to validate the WRF-Chem model performance in simulating air
pollutants and aerosol species against observations and measurements. IOA describes the
relative difference between the model predictions and observations, ranging from 0 to 1, with
1 indicating perfect agreement of predictions and observations.
$$MB = \frac{1}{N}\sum_{i=1}^{N}(P_i - O_i)$$
$$RMSE = \left[\frac{1}{N}\sum_{i=1}^{N}(P_i - O_i)^2\right]^{\frac{1}{2}}$$
$$IOA = 1 - \frac{\sum_{i=1}^{N}(P_i - O_i)^2}{\sum_{i=1}^{N}(|P_i - \overline{O}| + |O_i - \overline{O}|)^2}$$
Where $P_i$ and $O_i$ are the predicted and observed mass concentrations of pollutants,
respectively. N is the total number of the predictions used for comparisons, and $\overline{P}$ and $\overline{O}$
represent the average of predictions and observations, respectively.

## 2.3 Pollutants measurements

The hourly near-surface CO, $SO_2$, $NO_2$, $O_3$, and $PM_{2.5}$ mass concentrations released by the
China's Ministry of Environmental Protection can be downloaded from the website
http://www.aqistudy.cn/. The sulfate, nitrate, ammonium, and organic aerosols (OA) have been
measured by the Aerodyne High Resolution Time-of-Flight Aerosol Mass Spectrometer (HR-
ToF-AMS) with a novel $PM_{2.5}$ lens from 9 to 26 January 2014 at the Institute of Remote
Sensing and Digital Earth (IRSDE), Chinese Academy of Sciences (40.00°N, 116.38°E) in
Beijing (Fig. 1) (Williams et al., 2013). The Positive Matrix Factorization (PMF) technique is
used with constraints implemented in SoFi (Canonaco et al., 2013) to analyze the sources of
OA and five components are separated by their mass spectra and time series. The components
include hydrocarbon-like OA (HOA), cooking OA (COA), biomass burning OA (BBOA), coal
combustion OA (CCOA), and oxygenated OA (OOA). HOA, COA, BBOA, and CCOA are
interpreted for surrogates of primary OA (POA), and OOA is a surrogate for SOA. Detailed
information about the HR-ToF-AMS measurements and data analysis can be found in Elser et
al. (2016).

## 3 Results and discussions

### 3.1 Model performance

#### 3.1.1 Air pollution simulations in BTH

Considering the key role of meteorological fields in determining the formation
transformation, diffusion, transport, and removal of the air pollutants, Figure 2 presents the
diurnal profiles of the observed and simulated temperature, relative humidity (RH), wind speed
and direction at meteorological sites in Beijing, Tianjin, and Shijiazhuang during the simulation

period. The WRF-Chem model reasonably well predicts the diurnal variations of the temperature in the three cities against observations, with IOAs of around 0.80. The model also well yields the temporal variation of the RH in Beijing when compared with observations, but it tends to underestimate the RH in Tianjin and Shijiazhuang with IOAs less than 0.70, and generally fails to capture the high RH exceeding 80%. The temporal variations of the wind speed and direction in BTH are also reasonably reproduced, but the model biases are still rather large.

Figure 3 presents the distributions of predicted and observed near-surface mass concentrations of $PM_{2.5}$, $O_3$, $NO_2$, and $SO_2$ along with the simulated wind fields averaged from 9 to 25 January 2014 in BTH. Generally, the predicted spatial pattern of $PM_{2.5}$ is well consistent with observations at ambient monitoring sites in BTH. The WRF-Chem model reasonably reproduces the high $PM_{2.5}$ concentrations exceeding 150 µg m$^{-3}$ in the plain region of BTH. Apparently, during the simulation period, the weak winds in the plain region of BTH facilitate the accumulation of $PM_{2.5}$, causing severe air pollution. The average simulated $PM_{2.5}$ mass concentrations exceed 250 µg m$^{-3}$ in south Hebei, which is generally in good agreement with observations. The observed and simulated $O_3$ concentrations are rather low in the plain region of BTH with the high $PM_{2.5}$ level, varying from 10 to 30 µg m$^{-3}$. There are several reasons for the low $O_3$ concentrations in the plain region of BTH. Firstly, during wintertime, the insolation is weak in north China, which is unfavorable for the $O_3$ photochemical production. Additionally, high $PM_{2.5}$ concentrations and frequent occurrence of clouds during haze days further attenuate the incoming solar radiation in the planetary boundary layer (PBL), decreasing the $O_3$ levels. Secondly, weak winds indicate stagnant situations, lacking the $O_3$ transport from outside BTH. Thirdly, high $NO_x$ emissions cause titration of $O_3$, which is shown by the high $NO_2$ concentrations in the plain region of BTH. The elevated $NO_2$ and $SO_2$ concentrations are observed and simulated in the plain region of BTH, particularly in cities and their surrounding

areas, ranging from 50 to 100 µg m$^{-3}$ and 50 to 150 µg m$^{-3}$, respectively. It is worth noting that
the simulated NO$_2$ is generally distributed evenly in the plain region of BTH, indicating the
dominant contribution of area sources.
Figure 4 presents the diurnal profiles of observed and simulated near-surface PM$_{2.5}$, O$_3$,
NO$_2$, SO$_2$, and CO mass concentrations averaged over all monitoring sites in BTH from 9 to
25 January 2014. The WRF-Chem model reproduces the diurnal variations of PM$_{2.5}$ mass
concentrations when compared with observations in BTH during the simulation period. The
MB and RMSE are only -2.7 and 40.9 µg m$^{-3}$, respectively, and the IOA is 0.94. During the
persistent haze episode in BTH, the model generally well replicates the haze developing stage,
but tends to underestimate the PM$_{2.5}$ concentrations against observations during the haze
dissipation stage. One of the most possible reasons is the uncertainty of the simulated
meteorological fields, which determine the formation, transformation, diffusion, transport, and
removal of air pollutants in atmosphere (Bei et al., 2012, 2013). Should the predicted winds be
intensified earlier than observations in BTH during the haze dissipation stage, the simulated
PM$_{2.5}$ concentrations would decline earlier, causing the model underestimation. The predicted
NO$_2$ diurnal variations are generally well consistent with observations, with a MB of 4.2 µg m$^{-}$
$^3$ and an IOA of 0.93. The model also yields reasonable predictions for SO$_2$ and CO temporal
variations with IOAs exceeding 0.85. However, the RMSE for SO$_2$ is rather large, showing
considerable deviations of the SO$_2$ simulations. A large fraction of SO$_2$ are emitted from power
plants or agglomerated industrial zones, which can be regarded as point sources, so the
transport of SO$_2$ is more sensitive to uncertainties in simulated wind fields. The early
occurrence of intensified winds in simulations also cause rapid falloff of SO$_2$ and CO mass
concentrations during the haze dissipation stage. Besides uncertainties in meteorological field
simulations, uncertainties in emission inventory are also responsible for the model biases of air
pollutants. Since implementation of the APPCAP, strict emission control measures have been

made to improve the air quality in BTH, and the spatiotemporal variations of anthropogenic emissions in BTH have changed considerably (Li et al., 2017), which is not reflected in the emission inventory used in the present study.

Recently, observational studies have used CO as an aerosol proxy to investigate atmospheric aerosols based on the remote sensing technique. Figure 5 shows the scatter plots of observed and simulated $PM_{2.5}$ with CO mass concentrations averaged over all ambient monitoring sites in BTH during the simulation period. The observed and simulated CO mass concentrations are well correlated with those of $PM_{2.5}$, with the $R^2$ exceeding 0.81.

Table 2 presents the further validation of WRF-Chem model simulations of air pollutants based on statistics methods suggested by previous studies (US EPA, 2005; Boylan and Russell, 2006; Emery et al., 2017). Compared to the suggested model performance criteria of air pollutants, the WRF-Chem model performs well in simulating the air pollutants and aerosol species in this study. The FB, FE, NMB, and NME of $PM_{2.5}$ and $O_3$ are generally within the benchmarks, with the correlation coefficients approaching 0.90, showing good consistency between the simulations and observations. As for the aerosol species, except for sulfate, the differences between the observed and simulated organic aerosol, nitrate, and ammonium are all less than the reference criteria. The FB and FE of sulfate are reasonable, but the NMB of 37.6% and NME of 67.8% are slightly higher than the suggested criteria.

### 3.1.2 Aerosol species simulations in Beijing

Figure 6 presents the temporal profiles of measured and simulated OA, CCOA, sulfate, nitrate, and ammonium mass concentrations at IRSDE site in Beijing from 9 to 25 January 2014. The model generally performs reasonably well in simulating the diurnal variations of aerosol species when compared with the HR-ToF-AMS measurements, with IOAs exceeding 0.80. As a primary species, OA is primarily determined by direct emissions from various sources, including vehicles, cooking, biomass burning, coal combustion, and transport from

outside Beijing. Therefore, uncertainties in anthropogenic emissions and the simulated meteorological fields markedly influence the OA simulations (Bei et al., 2017). Although the IOA for OA is 0.84, the model slightly overestimates the OA mass concentrations with a MB of 5.1 µg m$^{-3}$, and the deviation of OA simulations is also large, with a RMSE of 42.3 µg m$^{-3}$. In addition, the model fails to reproduce the measured OA peaks during the nighttime on 11 and 17 January 2014, which is perhaps caused by the emission uncertainties. The model also generally tracks the measured diurnal variations of CCOA mass concentrations, with an IOA of 0.81. The model frequently underestimates or overestimates the CCOA mass concentrations and is also subject to missing the observed CCOA peaks. The CCOA is mainly emitted from industries and residential coal combustion. In general, the CCOA emissions from industries have clear diurnal variations, but are opposite for those from residential coal combustion, causing large model biases for the CCOA simulation. The simulated time-series of sulfate, nitrate, and ammonium are also in good agreement with observations, with IOA of 0.83, 0.87, and 0.90, respectively. The model considerably overestimates the inorganic aerosol mass concentrations from 16 to 18 January. One of the possible reasons is the decreased emissions, particularly from industries before the Chinese New Year, which are not reflected in the emission inventory used in the study.

Figure 7 presents the contributions of aerosol species to the simulated PM$_{2.5}$ concentration in BTH and Beijing averaged from 9 to 25 January 2014. The modeled PM$_{2.5}$ mass concentration averaged during the simulation period in BTH and Beijing is 111.6 and 97.7 µg m$^{-3}$, respectively. OA dominate the PM$_{2.5}$ in BTH, with a contribution of around 43.1%. Although the simulated O$_3$ concentration is low, the secondary aerosols, including SOA, sulfate, nitrate, and ammonium still make up about 40% of the PM$_{2.5}$ mass concentration, with contributions of 7.9%, 11.3%, 12.4%, and 9.6%, respectively. Elemental carbon and the unspecified aerosol species account for 7.5% and 16.2% of the PM$_{2.5}$ mass concentration,

respectively. In Beijing, sulfate, nitrate, and ammonium constitutes 10.6%, 14.0%, and 9.1% of the $PM_{2.5}$ mass concentrations, respectively. OA are also the dominant constituent of the simulated $PM_{2.5}$ in Beijing, with a contribution of about 44.1%. The simulated ratio of the primary to secondary OA in Beijing is 4.6, which is close to the observed ratio of 4.3. The simulated chemical composition in Beijing is generally comparable to the observation in January 2013 by Huang et al. (2014), showing that OA constitutes a major fraction (40.7%) of the total $PM_{2.5}$, followed by sulfate (16.0%), nitrate (12.0%), and ammonium (9.8%). It is worth noting that the simulated sulfate contribution to $PM_{2.5}$ mass concentrations in Beijing is lower than the observation in Huang et al. (2014), and vice versa for the nitrate aerosol. Implementation of the APPCAP since 2013 September has considerably decreased $SO_2$ emissions in BTH, lowering the sulfate formation. Additionally, the decrease of the sulfate aerosol reduces its competition with ammonia in the atmosphere, facilitating the nitrate formation.

The good agreements of the simulated mass concentrations of air pollutants with observations at ambient monitoring sites in BTH and aerosol species with HR-ToF-AMS measurements in Beijing show that the simulated wind fields and emission inventory used in present study are generally reasonable, providing a reliable base for further evaluations.

**3.2   Contributions of the RCC emission to the air quality in BTH**

The contribution of the residential coal combustion (RCC) emission to the air quality in BTH is investigated by the sensitivity study without RCC emissions in BTH and its surrounding areas compared to the reference simulation. Figure 8 shows the spatial distribution of the average contribution of the RCC emission in BTH to $PM_{2.5}$ mass concentrations during the simulation period (REF - SEN-BTH). The RCC emission plays an important role in the $PM_{2.5}$ level in the plain area of BTH, with contributions varying from 30 to 70 µg m$^{-3}$. Over the mountain areas of BTH, the contribution of RCC emissions to the $PM_{2.5}$ mass concentration

is generally less than 10 $\mu g\ m^{-3}$.
Table 3 presents the average change of air pollutants mass concentrations during the
simulation period in BTH and Beijing. The average $PM_{2.5}$ mass concentration is 111.6 $\mu g\ m^{-3}$
in BTH in REF case and decreased to be 85.8 $\mu g\ m^{-3}$ in SEN-BTH case when the RCC emission
in BTH is excluded. The RCC emission contributes about 23.1% of $PM_{2.5}$ mass concentrations
in BTH on average. In addition, the RCC emission is an important source of $SO_2$ and CO,
contributing about 35.8% of $SO_2$ and 22.5% of CO mass concentrations. The RCC emission
does not substantially influence the $NO_2$ level in BTH, with a contribution of 4.2%. When the
RCC emission in BTH is not considered, the $O_3$ concentration slightly increases due to the
decrease of $NO_2$ concentration. The $PM_{2.5}$ mass concentration is decreased by around 30% in
Beijing on average when the RCC emission in BTH is excluded, showing that the air quality
in Beijing would be remarkably improved if the residential coal in BTH and its surrounding
areas could be replaced by other clean energy sources, such as natural gas or electricity.
Furthermore, the RCC emission in BTH contributes about 42.6% of $SO_2$ and 26.5% of CO
mass concentrations in Beijing.
Figure 9 shows the average chemical composition of $PM_{2.5}$ contributed by the RCC
emission in BTH and Beijing during the simulation period. The RCC emission contributes
about 25.8 $\mu g\ m^{-3}$ $PM_{2.5}$ in BTH on average, of which about 42.8% is from OA. The sulfate
aerosol constitutes 17.1% of the $PM_{2.5}$ from the RCC emission, exceeding the contribution
from unidentified aerosol species (15.8%), element carbon (11.5%), ammonium (9.5%) and
nitrate (3.3%) aerosol. The results indicate that the priority to mitigate effects of the RCC
emission on the air quality in BTH is to decrease the emissions of OA and $SO_2$ from RCC. In
Beijing, OA is still the major contributor to $PM_{2.5}$ from the RCC emission, accounting for about
48.5%, which is more than that averaged in BTH. The sulfate and ammonium contribution to
the $PM_{2.5}$ from the RCC emission is 13.3% and 7.2%, respectively. The chemical composition

of the PM$_{2.5}$ from the RCC emission in Beijing shows more contribution of OA and less contribution of SO$_2$ from Beijing local RCC emission. It is worth noting that light absorbing aerosols are thought to alter the ambient temperature profile locally (Wang et al., 2013; Zhang et al., 2015; Peng et al., 2016). The sensitivity results indicate that if the RCC emission in BTH and its surrounding areas is excluded, the surface temperature in BTH is decreased by about 0.23℃ on average during the study period, about half of which is contributed by light absorbing aerosols.

**3.3    Contributions of local RCC emission to the air quality in Beijing**

As the capital of China, the air quality in Beijing often becomes the focus of attention in China or globally. Beijing is situated at the northern tip of the North China Plain (NCP), one of the most polluted areas in China, caused by rapid industrialization and urbanization (Zhang et al., 2013). In addition, Beijing is surrounded from southwest to northeast by the Taihang Mountains and the Yanshan Mountains which block the dispersion of air pollutants when south or east winds are prevalent in NCP (Long et al., 2016). Therefore, in addition to the contribution of local emissions, the air quality in Beijing is also substantially influenced by the transport of air pollutants from outside (Wu et al., 2017).

Since implementation of the APPCAP issued in September 2013, Beijing has carried out aggressive emission control strategies to improve air quality. Great efforts have been made to replace coal used in residential living by natural gas or electricity, which is highly anticipated to clean the air in Beijing. However, frequent occurrence of heavy haze with extremely high levels of PM$_{2.5}$ during the wintertime of 2015 and 2016 has caused controversial issue about the effect of the coal replacement plan in Beijing. Therefore, a further sensitivity study has been performed in this study, in which only the RCC emission in Beijing is excluded (SEN-PEK) to explore the contribution of the local RCC emission in Beijing to the haze formation. Comparisons of the REF case with the SEN-PEK case show that when the RCC emission in

Beijing is not considered or the residential coal is replaced by other clean energy sources, the local $PM_{2.5}$ level decreases from 97.7 to 80.1 μg m$^{-3}$ or by 18.0% on average during the simulation period. The average decreases in $SO_2$ and CO concentrations are 24.2% and 19.9%, respectively. It is worthy to note that the electricity is principally from the coal burning in China, and the main air pollutants emitted from coal-burning power plants are $NO_x$ and $SO_2$. However, the major pollutants emitted by the residential coal combustion include organic carbon, $SO_2$ and $NO_x$. Considering the dominant role of OA in the $PM_{2.5}$ in Beijing, the coal replacement in residential living is more effective in power plants. Therefore, the coal replacement plan in Beijing can improve the local air quality considerably, but is not as expected to substantially improve the air quality.

It is still debatable on whether local emissions or transport dominates the air quality in Beijing (Guo et al., 2010, 2014; Li et al., 2015; Zhang et al., 2015; Wu et al., 2017). Sensitivity studies show that when only the RCC emission in Beijing is excluded in simulations, the $PM_{2.5}$ level is decreased by 18%, much less than about 30% decrease caused by the exclusion of the RCC emission in BTH and its surrounding areas, showing the important contribution of trans-boundary transport to the air quality in Beijing. Analyses are further made to examine the contribution of the RCC emission in Beijing to the $PM_{2.5}$ mass concentrations under different pollution levels. The simulated hourly near-surface $PM_{2.5}$ mass concentrations in REF case during the whole episode in Beijing are first subdivided into 6 bins according to the air quality standard in China for $PM_{2.5}$ (Feng et al., 2016), i.e., 0~35 (excellent), 35~75 (good), 75~115 (lightly polluted), 115~150 (moderately polluted), 150~250 (heavily polluted), and greater than 250 (severely polluted) μg m$^{-3}$. $PM_{2.5}$ mass concentrations in REF case and SEN-PEK case as the bin $PM_{2.5}$ concentrations in REF case following the grid cells are assembled respectively, and an average of $PM_{2.5}$ mass concentrations in each bin is calculated. Figure 10 presents the contribution of the RCC emission in Beijing to the local $PM_{2.5}$ mass concentrations. Apparently,

the mitigation effect is the best under good and lightly polluted conditions in terms of $PM_{2.5}$ level, and the $PM_{2.5}$ mass concentration decreases by around 25% when the RCC emission in Beijing is not considered, indicating that the local RCC emission does not constitute the main $PM_{2.5}$ pollution source in Beijing. However, with the deterioration of haze pollution from moderately to severely polluted conditions, the $PM_{2.5}$ contribution from the local RCC emission in Beijing decreases from 20% to 15%, showing the regional transport of $PM_{2.5}$.

## 4 Summary and Conclusions

In the present study, a persistent air pollution episode in BTH from 9 to 25 January 2014 is simulated using the WRF-Chem model to assess contributions of the RCC emission to the air quality in BTH. In general, the WRF-Chem model performs well in simulating the temporal variations and spatial distributions of air pollutants when compared with observations over monitoring sites in BTH. The simulated diurnal variations of aerosol species are also in good agreements with the HR-ToF-AMS measurements in Beijing.

Sensitivity studies show that, on average, the RCC emission contributes about 23.1% of $PM_{2.5}$ mass concentrations in BTH during the simulation period and is also an important $SO_2$ and CO source, accounting for about 35.8% of $SO_2$ and 22.5% of CO mass concentrations. OA is the major contributor to $PM_{2.5}$ from the RCC emission, with a contribution of 42.8%, followed by sulfate (17.1%), unspecified species (15.8%), element carbon (11.5%), ammonium (9.5%) and nitrate (3.3%) aerosol. Exclusion of the RCC emission in BTH decreases the $PM_{2.5}$ concentration by around 30% in Beijing, indicating that the air quality in Beijing will be remarkably improved if the residential coal in BTH and its surrounding areas can be replaced by other clean energy sources.

When only the RCC emission in Beijing is excluded in simulations, Beijing's $PM_{2.5}$ level decreases by 18.0% on average during the simulation period. Hence, the coal replacement plan

in Beijing is beneficial to the local air quality, but is not as anticipated to substantially improve the air quality. The mitigation effect of the coal replacement plan on $PM_{2.5}$ in Beijing is the best under good and lightly polluted conditions, decreasing the $PM_{2.5}$ mass concentration by around 25%. However, under heavy or severe haze pollution, the local RCC emission contributes about 15% of $PM_{2.5}$ in Beijing, showing the regional transport of $PM_{2.5}$.

This study mainly aims to quantitatively evaluate the contributions of the RCC emission to the air quality in BTH. Our results indicate that if the residential coal replacement is only implemented in Beijing, Beijing's air quality will be improved considerably, but not substantially, considering the impact of trans-boundary transport. Implementation of the residential coal replacement in BTH and its surrounding areas can remarkably improve Beijing's air quality. Although the WRF-Chem model reasonably captures the temporal and spatial variations of air pollutants in BTH and diurnal variations of aerosol species in Beijing, the model biases still exit. Future studies need to be conducted to improve the WRF-Chem model simulations, considering the rapid changes in anthropogenic emissions since the implementation of APPCAP. Further sensitivity simulations of various emission mitigation measures also need to be performed to provide efficient emission control strategies to improve the air quality in BTH.

## 5 Data availability

The real-time $O_3$ and $PM_{2.5}$ mass concentrations are accessible to the public on website http://106.37.208.233:20035/ (China MEP, 2013a). One can also access the historic profiles of the observed ambient air pollutants by visiting http://www.aqistudy.cn/ (China MEP, 2013b).

Acknowledgements. This work is financially supported by the National Key R&D Plan (Quantitative Relationship and Regulation Principle between Regional Oxidation Capacity of

441 Atmospheric and Air Quality (2017YFC0210000)). Guohui Li is supported by "Hundred

Talents Program" of the Chinese Academy of Sciences and the National Natural Science

Foundation of China (No. 41661144020).

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

Table 1 WRF-Chem model configurations.

| Region | Beijing-Tianjin-Hebei (BTH) |
|---|---|
| Simulation period | January 9 to 26, 2014 |
| Domain size | 150 × 150 |
| Domain center | 39°N, 117°E |
| Horizontal resolution | 6km × 6km |
| Vertical resolution | 35 vertical levels with a stretched vertical grid with spacing ranging from 30 m near the surface, to 500 m at 2.5 km and 1 km above 14 km |
| Microphysics scheme | WSM 6-class graupel scheme (Hong and Lim, 2006) |
| Boundary layer scheme | MYJ TKE scheme (Janjić, 2002) |
| Surface layer scheme | MYJ surface scheme (Janjić, 2002) |
| Land-surface scheme | Unified Noah land-surface model (Chen and Dudhia, 2001) |
| Longwave radiation scheme | Goddard longwave scheme (Chou and Suarez, 2001) |
| Shortwave radiation scheme | Goddard shortwave scheme (Chou and Suarez, 1999) |
| Meteorological boundary and initial conditions | NCEP 1°×1° reanalysis data |
| Chemical initial and boundary conditions | MOZART 6-hour output (Horowitz et al., 2003) |
| Anthropogenic emission inventory | Developed by Zhang et al. (2009) and Li et al. (2017) |
| Biogenic emission inventory | MEGAN model developed by Guenther et al. (2006) |


Table 2 Validation of WRF-Chem model performance on simulations of air pollutants.

| Species | FB[a] | | FE[b] | | NMB[c] | | NME[d] | | r[e] | |
|---|---|---|---|---|---|---|---|---|---|---|
| | This study | Ref. criteria | This study | Ref. criteria | This study | Ref. criteria | This study | Ref. criteria | This study | Ref. criteria |
| PM$_{2.5}$ | -0.7% | < ±60%[1] | 21.5% | <±75%[1] | -1.7% | <±30%[2] | 18.5% | <±50%[2] | 0.89 | >0.40[2] |
| O$_3$ | -28.0% | | 40.5% | | -11.3% | <±15%[3] | 24.7% | <±35%[3] | 0.91 | >0.50[3] |
| OA | 21.7% | < ±60%[1] | 51.9% | <±75%[1] | 8.6% | <±50%[2] | 44.5% | <±65%[2] | 0.74 | |
| CCOA | 39.0% | < ±60%[1] | 64.7% | <±75%[1] | 21.5% | <±50%[2] | 59.6% | <±65%[2] | 0.69 | |
| Sulfate | 46.5% | < ±60%[1] | 64.7% | <±75%[1] | 37.6% | <±30%[2] | 67.8% | <±50%[2] | 0.75 | >0.40[2] |
| Nitrate | 6.0% | < ±60%[1] | 56.2% | <±75%[1] | -1.3% | <±65%[2] | 46.5% | <±115%[2] | 0.78 | |
| Ammonium | 11.5% | < ±60%[1] | 46.3% | <±75%[1] | -2.7% | <±30%[2] | 37.4% | <±50%[2] | 0.83 | >0.40[2] |

[a] Fractional bias (FB): $FB = \frac{2}{N} \sum \frac{(P_j - O_j)}{(P_j + O_j)} \times 100$
[b] Fractional error (FE): $FE = \frac{2}{N} \sum \frac{|P_j - O_j|}{(P_j + O_j)} \times 100$
[c] Normalized mean bias (NMB): $NMB = \frac{\sum (P_j - O_j)}{\sum O_j} \times 100$
[d] Normalized mean error (NME): $NME = \frac{\sum |P_j - O_j|}{\sum O_j} \times 100$
[e] Correlation coefficient (r): $r = \frac{\sum [(P_j - \bar{P}) \times (O_j - \bar{O})]}{\sqrt{\sum (P_j - \bar{P})^2 \times \sum (O_j - \bar{O})^2}}$
Where subscript j represents the pairing of N, observations O, and predictions P, by site and time. r = 1 is
perfect correlation, r = 0 is totally uncorrelated.
[1] Boylan and Russell (2006)
[2] Emery et al. (2017)
[3] US EPA (2005)

Table 3 Average mass concentrations of air pollutants in REF case and SEN-BTH case from 9 to 25
January 2014 in BTH and Beijing. (Unit: μg m$^{-3}$ for $PM_{2.5}$, $O_3$, $NO_2$, $SO_2$ and mg m$^{-3}$ for CO)

| Air pollutants | BTH | | | | Beijing | | | |
|---|---|---|---|---|---|---|---|---|
| | REF | SEN-BTH | Mass change | Percentage change | REF | SEN-BTH | Mass change | Percentage change |
| $PM_{2.5}$ | 111.6 | 85.8 | 25.8 | 23.1% | 97.7 | 68.9 | 28.8 | 29.5% |
| $O_3$ | 39.1 | 39.4 | -0.3 | -0.8% | 39.3 | 39.8 | -0.5 | -1.3% |
| $NO_2$ | 45.7 | 43.7 | 2.0 | 4.3% | 51.5 | 49.4 | 2.1 | 4.1% |
| $SO_2$ | 45.0 | 28.9 | 16.1 | 35.8% | 42.2 | 24.2 | 18.0 | 42.6% |
| CO | 1.7 | 1.3 | 0.4 | 22.5% | 1.5 | 1.1 | 0.4 | 26.5% |


**Figure Captions**

Figure 1 (a) Map showing the location of Beijing-Tianjin-Hebei and (b) WRF-Chem model
simulation domain with topography. In (b), the filled red circles represent centers of cities
with ambient monitoring site and the size of the circle denotes the number of ambient
monitoring sites of cities. The filled black rectangle denotes the deployment location of the
HR-ToF-AMS in Beijing. The three filled black circles represent the location of the
meteorological observation stations in Beijing, Tianjin, and Shijiazhuang, respectively.

Figure 2 Comparisons of observed (black dots) and simulated (solid red lines) diurnal profiles of
near-surface temperature (T), relative humidity (RH), wind speed (WS), and wind direction
(WD) at meteorological sites in (a) Beijing, (b) Tianjin, and (c) Shijiazhuang from 9 to 25
January 2014.

Figure 3 Pattern comparisons of simulated (color counters) vs. observed (colored circles) near-
surface mass concentrations of (a) $PM_{2.5}$, (b) $O_3$, (c) $NO_2$, and (d) $SO_2$ averaged from 9 to
25 January 2014. The black arrows indicate simulated surface winds.

Figure 4 Comparisons of observed (black dots) and simulated (solid red lines) diurnal profiles of
near-surface hourly mass concentrations of (a) $PM_{2.5}$, (b) $O_3$, (c) $NO_2$, (d) $SO_2$, and (d) CO
averaged at monitoring sites in BTH from 9 to 25 January 2014.

Figure 5 Scatter plots of the (a) observed and (b) simulated $PM_{2.5}$ with CO mass concentrations
averaged over all ambient monitoring sites in BTH from 9 to 25 January 2014.

Figure 6 Comparisons of measured (black dots) and simulated (solid red lines) diurnal profiles of (a)
organic aerosol (OA), (b) coal combustion organic aerosol (CCOA), (c) sulfate, (d) nitrate,
and (e) ammonium in Beijing from 9 to 25 January 2014.

Figure 7 Chemical composition of $PM_{2.5}$ averaged from 9 to 25 January 2014 in (a) BTH and (b)
Beijing.

Figure 8 Spatial distribution of the average contribution of the RCC emission in BTH to $PM_{2.5}$ mass
concentrations from 9 to 25 January 2014.

Figure 9 Chemical composition of $PM_{2.5}$ from the RCC emission in BTH averaged from 9 to 25
January 2014 in (a) BTH and (b) Beijing.

Figure 10 Average contributions of the RCC emission in Beijing to the local $PM_{2.5}$ mass
concentrations under different haze pollution levels from 9 to 25 January 2014. *The green,*
*yellow, orange, red, purple, and dark red represents excellent, good, slightly polluted,*
*moderately polluted, heavily polluted, and severely polluted levels of air quality,*
*respectively.*

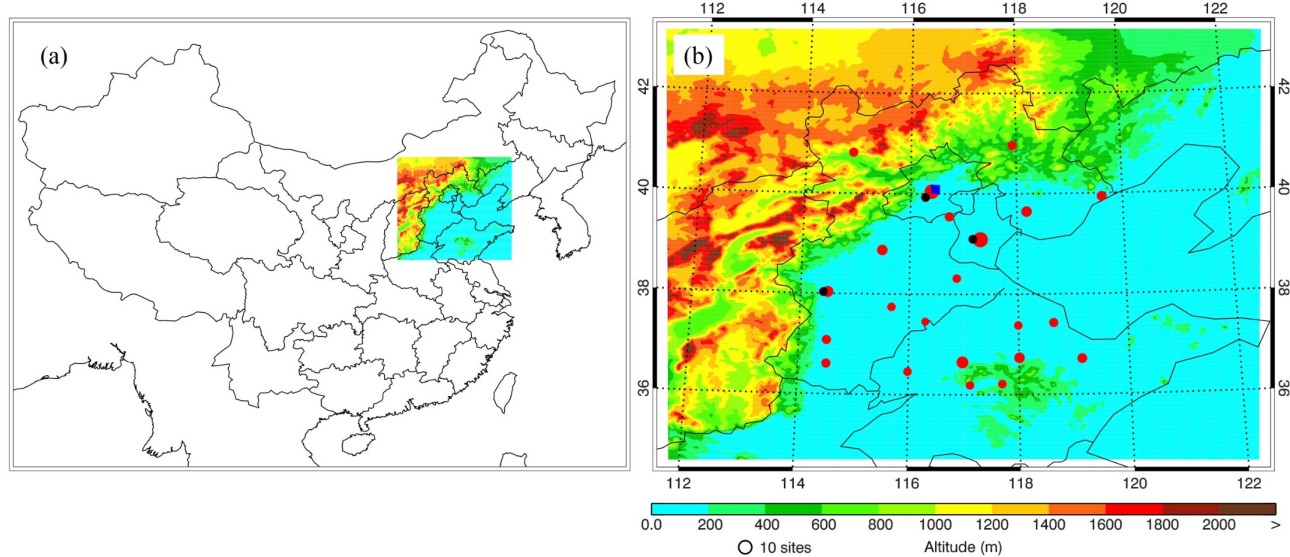

Figure 1 (a) Map showing the location of Beijing-Tianjin-Hebei and (b) WRF-Chem model
simulation domain with topography. In (b), the filled red circles represent centers of cities with
ambient monitoring site and the size of the circle denotes the number of ambient monitoring sites of
cities. The filled blue rectangle denotes the deployment location of the HR-ToF-AMS in Beijing. The
three filled black circles represent the location of the meteorological observation stations in Beijing,
Tianjin, and Shijiazhuang, respectively.

(a) Beijing

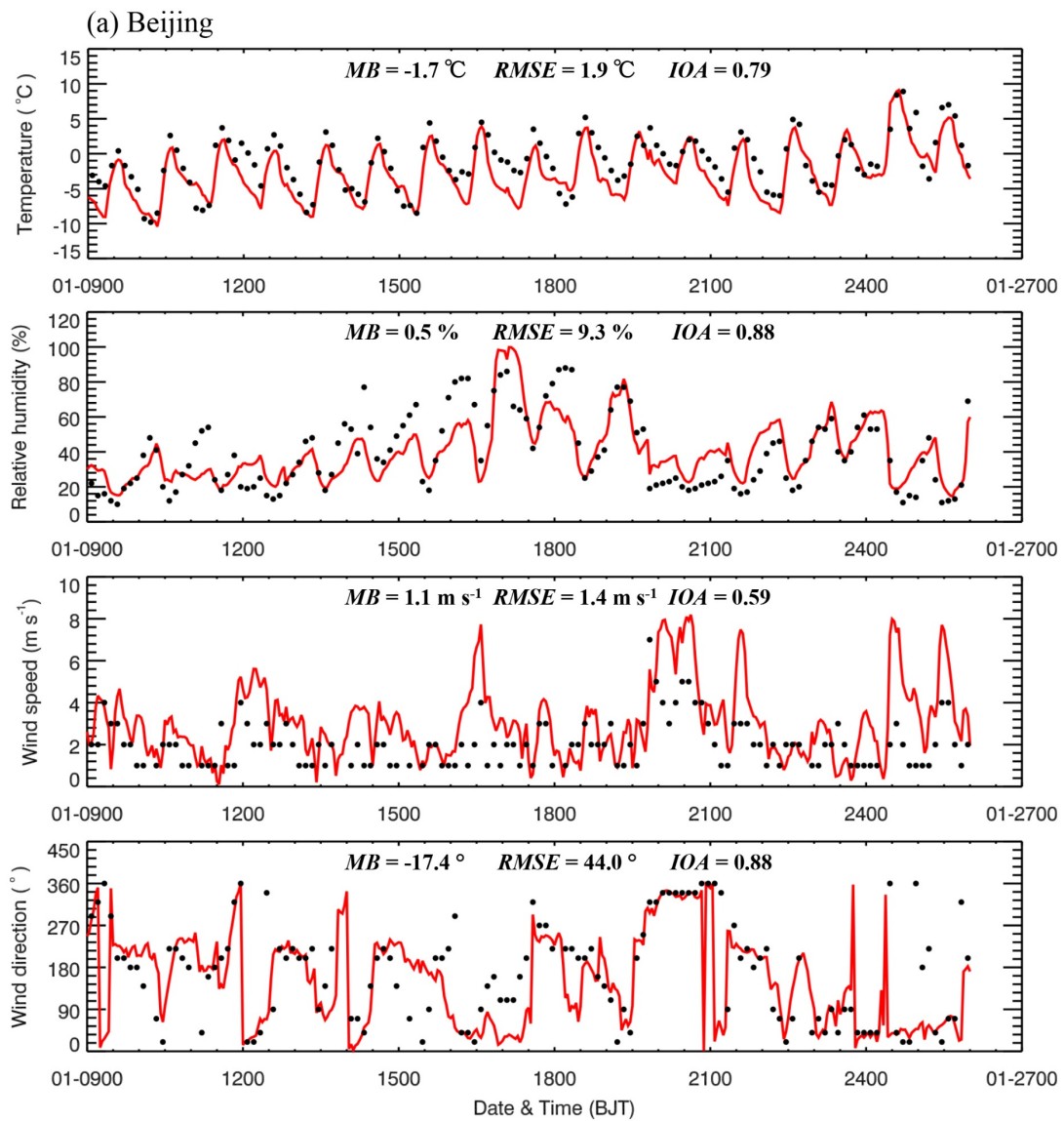

Figure 2 Comparisons of observed (black dots) and simulated (solid red lines) diurnal profiles of
near-surface temperature, relative humidity (RH), wind speed, and wind direction at meteorological
sites in (a) Beijing, (b) Tianjin, and (c) Shijiazhuang from 9 to 25 January 2014.

(b) Tianjin

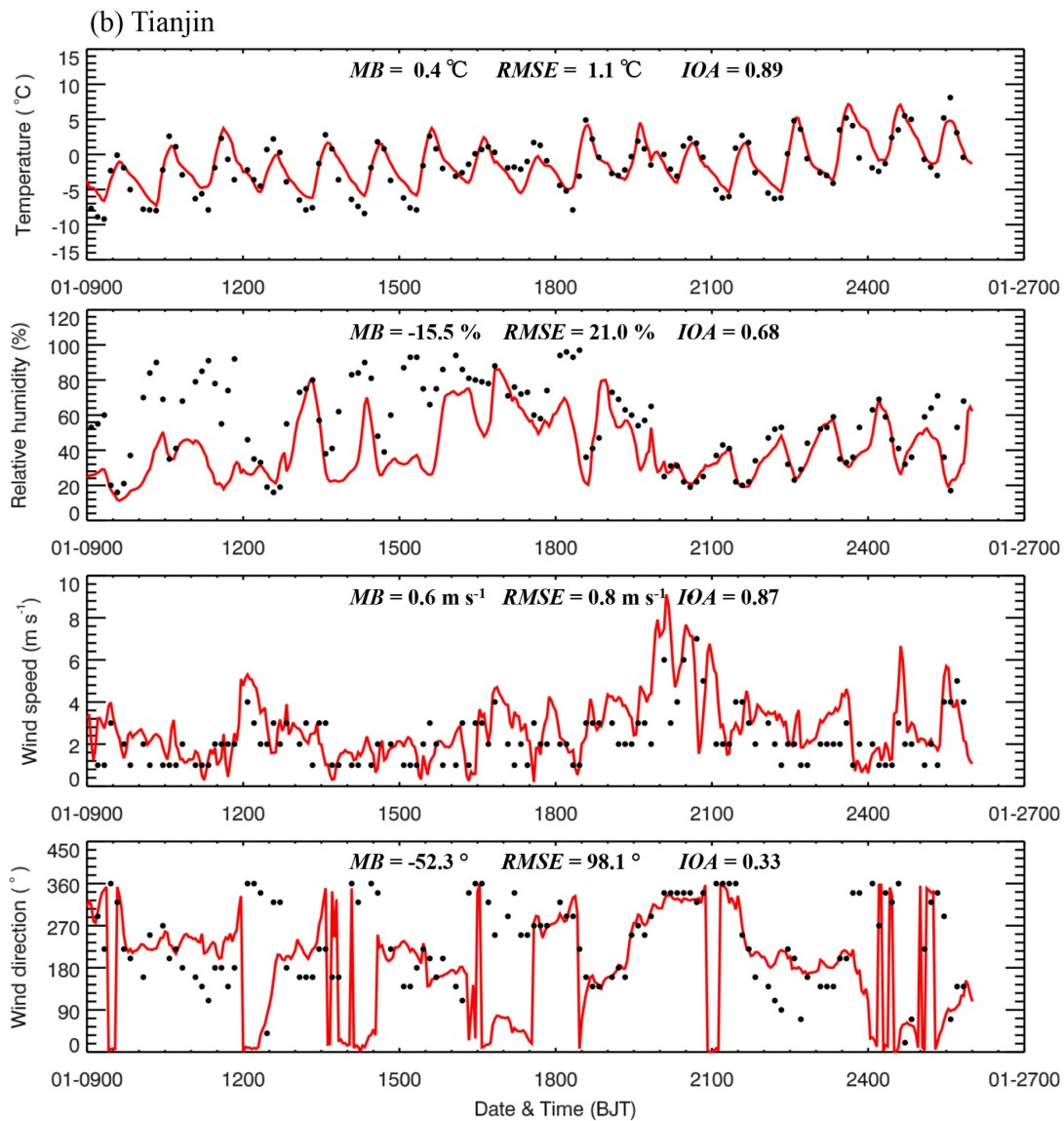

Figure 2 continued.

(c) Shijiazhuang

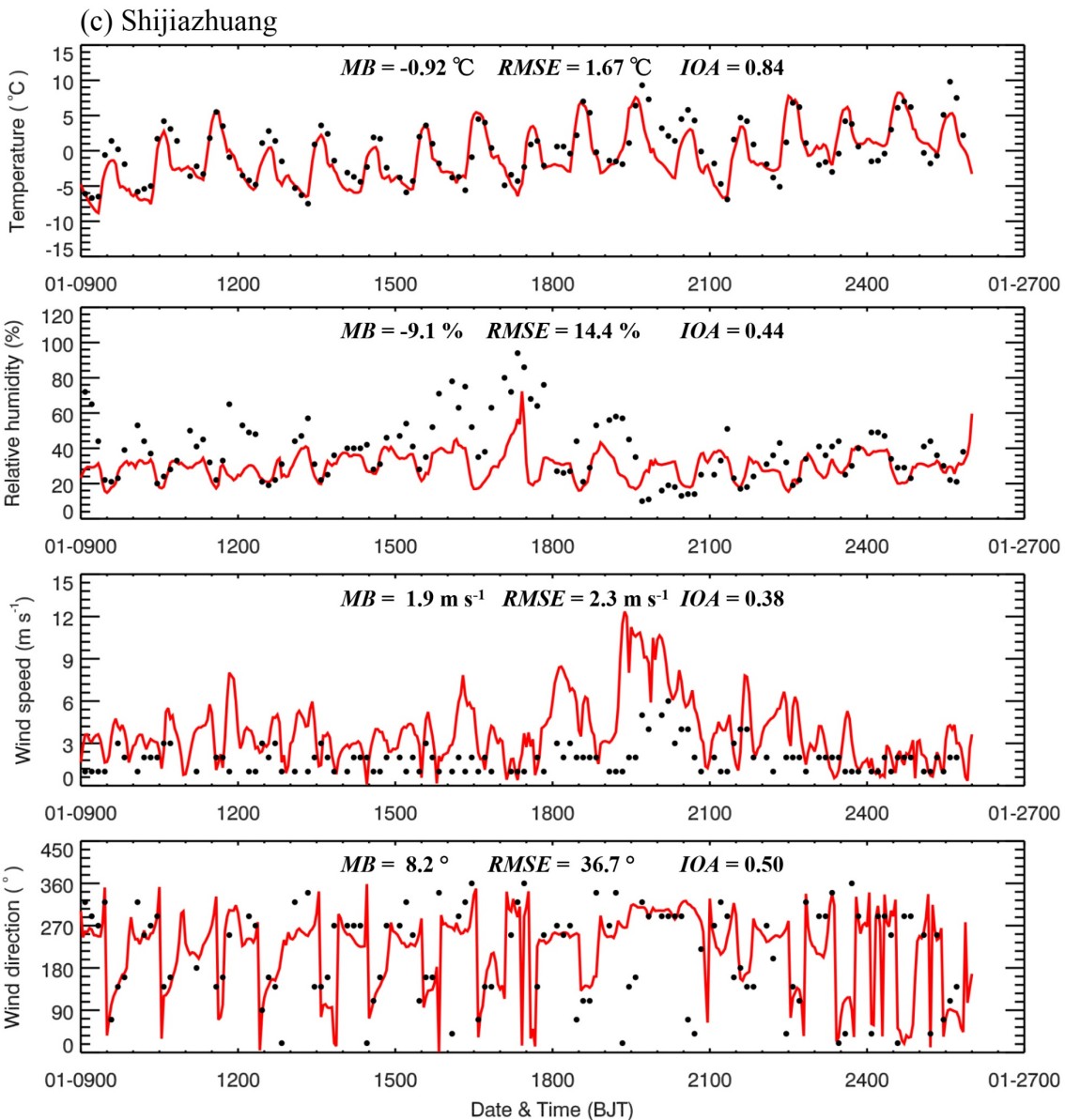

Figure 2 continued.

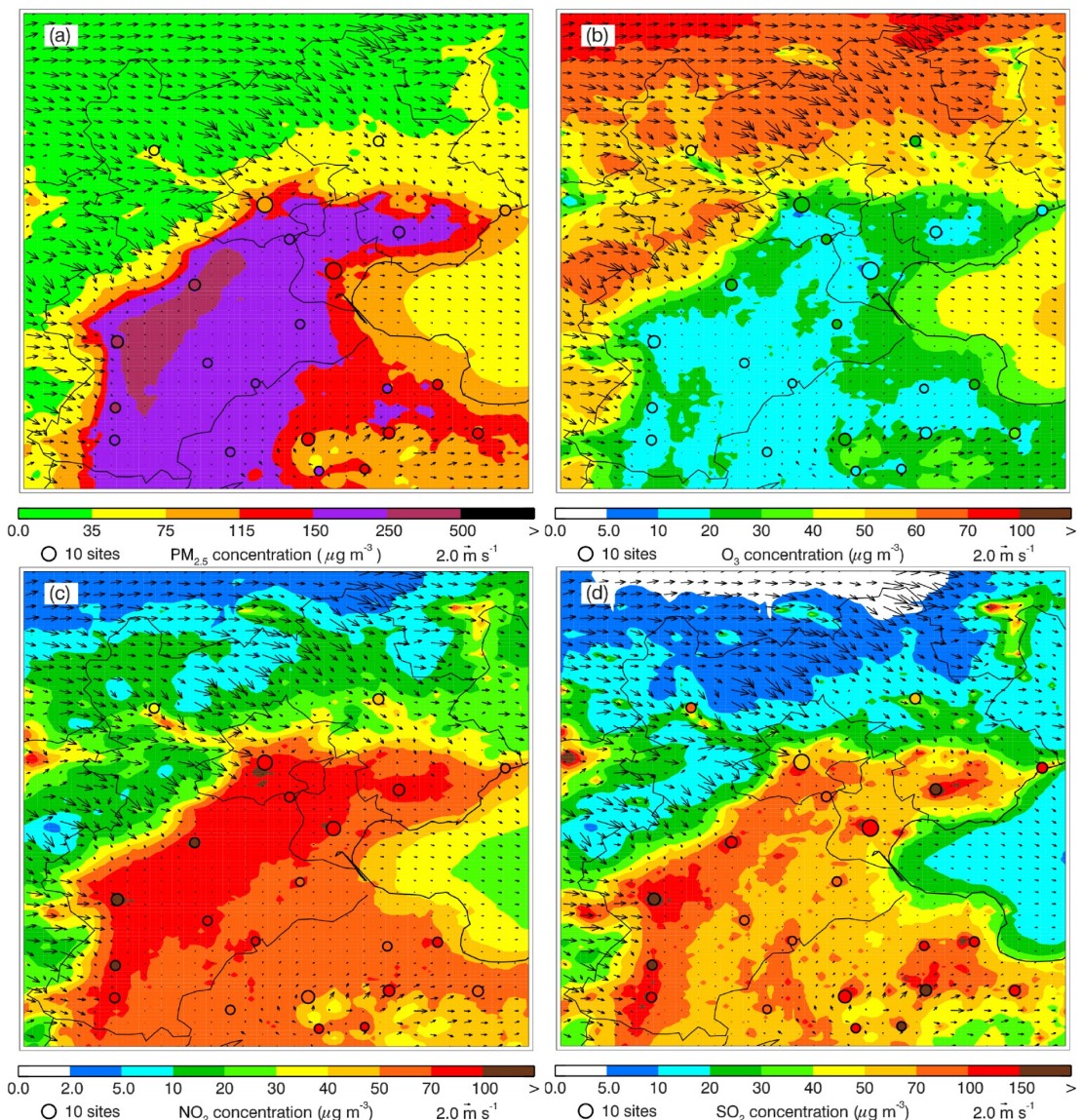

Figure 3 Pattern comparisons of simulated (color counters) vs. observed (colored circles) near-
surface mass concentrations of (a) $PM_{2.5}$, (b) $O_3$, (c) $NO_2$, and (d) $SO_2$ averaged from 9 to 25 January
2014. The black arrows indicate simulated surface winds.

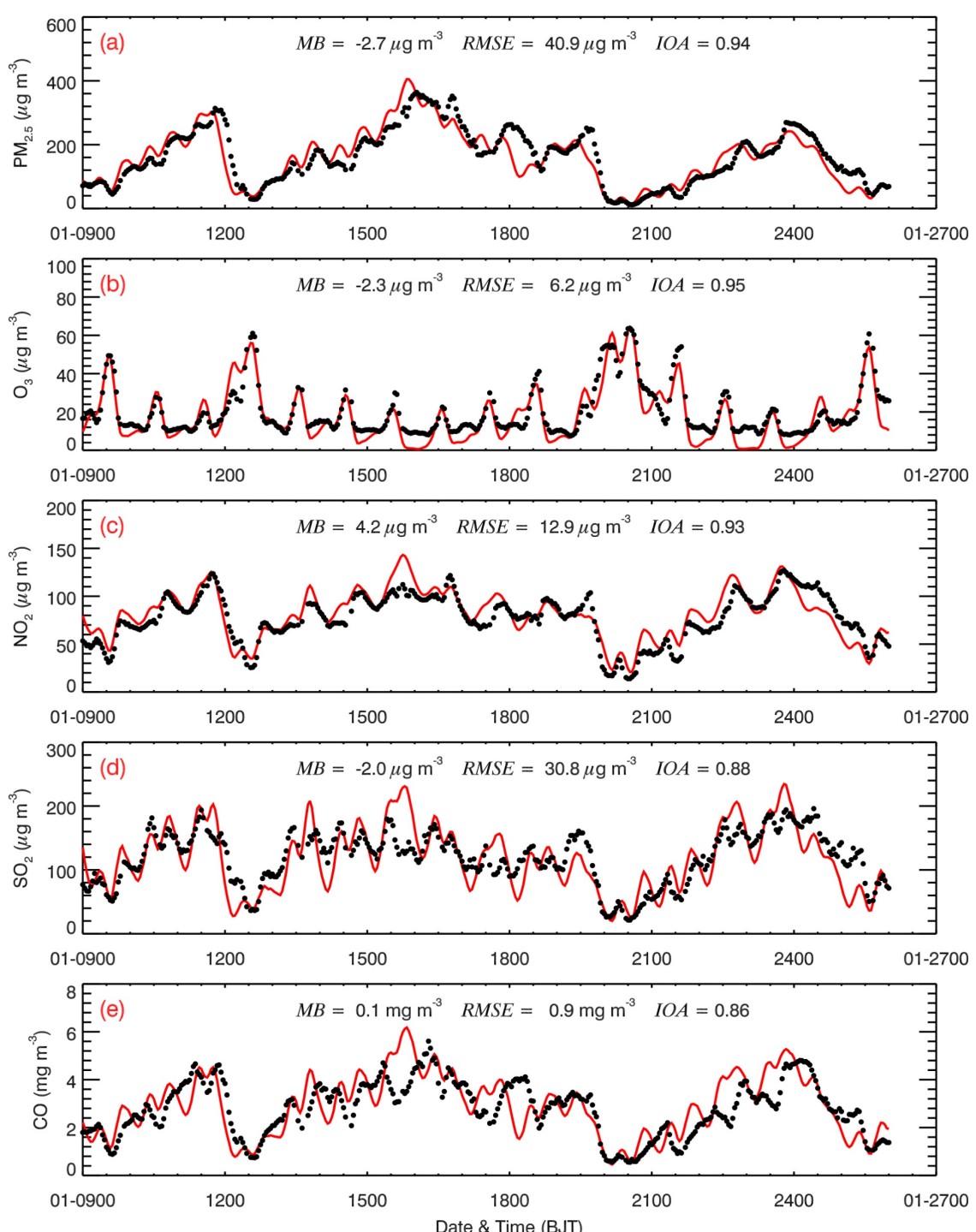

Figure 4 Comparisons of observed (black dots) and simulated (solid red lines) diurnal profiles of
near-surface hourly mass concentrations of (a) PM$_{2.5}$, (b) O$_3$, (c) NO$_2$, (d) SO$_2$, and (d) CO averaged
at monitoring sites in BTH from 9 to 25 January 2014.

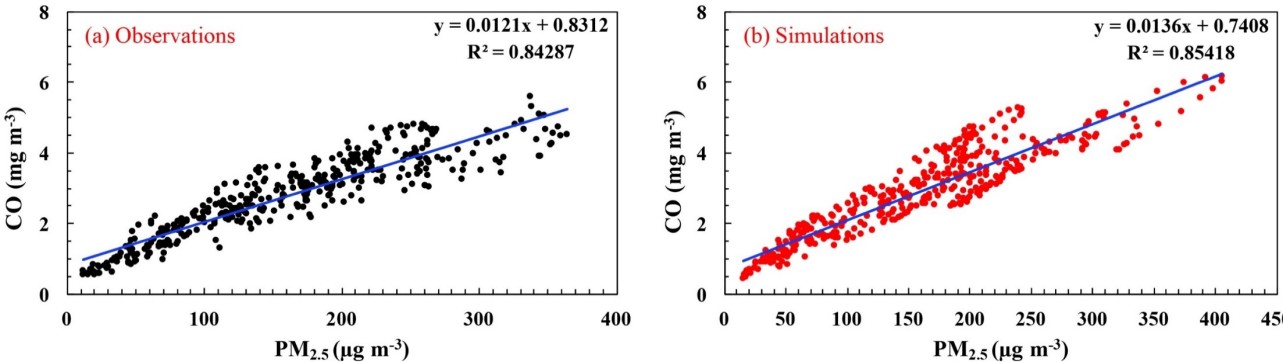



Figure 5 Scatter plots of the (a) observed and (b) simulated PM$_{2.5}$ with CO mass concentrations
averaged over all ambient monitoring sites in BTH from 9 to 25 January 2014.





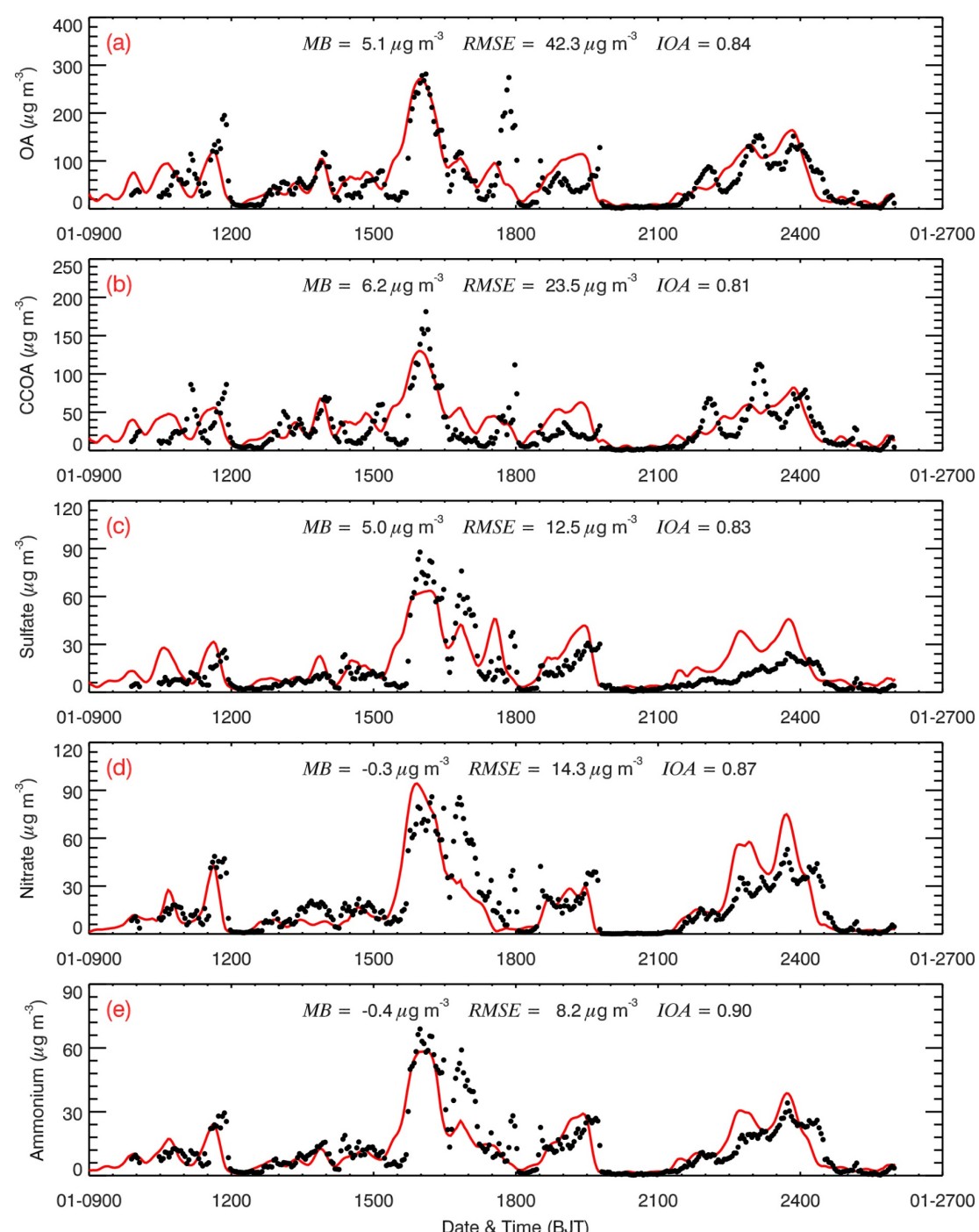

Figure 6 Comparisons of measured (black dots) and simulated (solid red lines) diurnal profiles of (a)
organic aerosol (OA), (b) coal combustion organic aerosol (CCOA), (c) sulfate, (d) nitrate, and (e)
ammonium in Beijing from 9 to 25 January 2014.

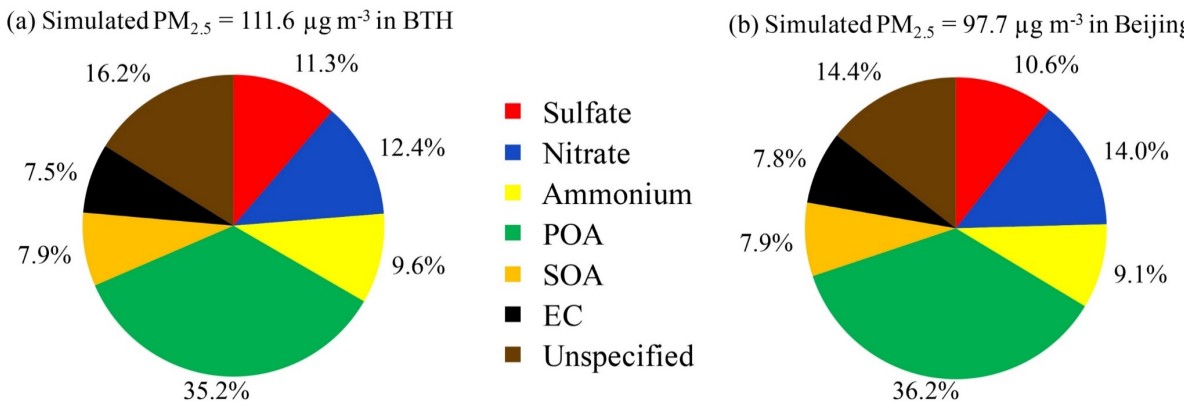

Figure 7 Chemical composition of PM$_{2.5}$ averaged from 9 to 25 January 2014 in (a) BTH and (b)
Beijing.

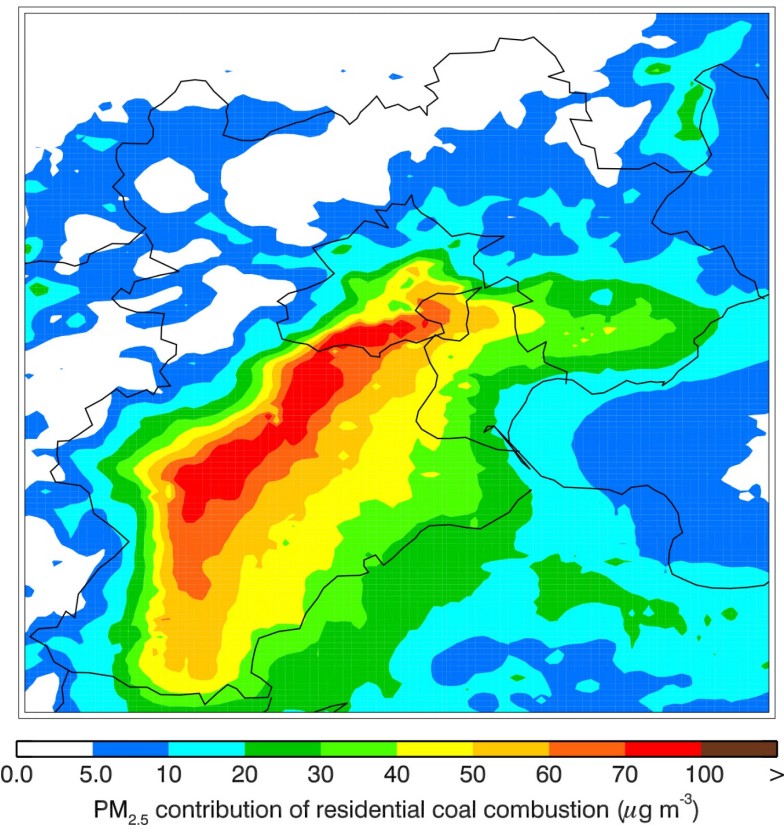

Figure 8 Spatial distribution of the average contribution of the RCC emission in BTH to PM$_{2.5}$ mass
concentrations from 9 to 25 January 2014.

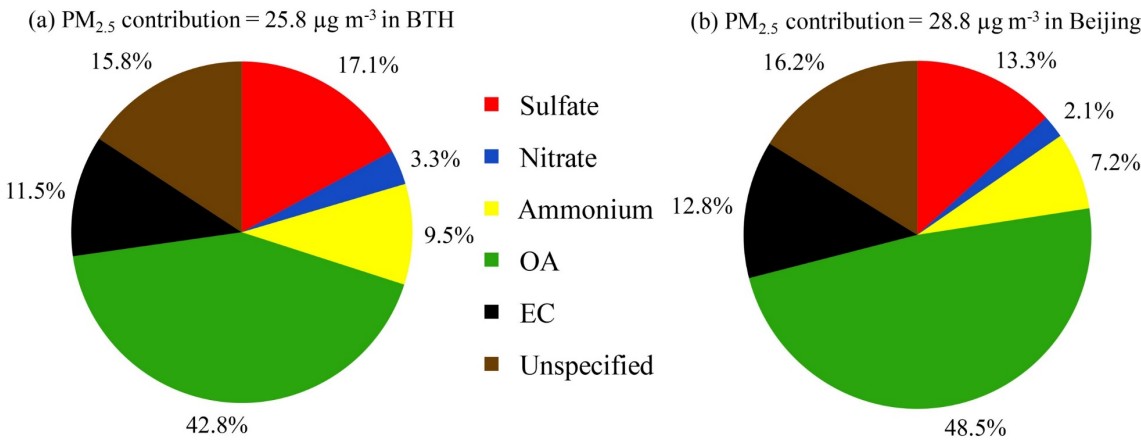

Figure 9 Chemical composition of PM$_{2.5}$ from the RCC emission in BTH averaged from 9 to 25
January 2014 in (a) BTH and (b) Beijing.

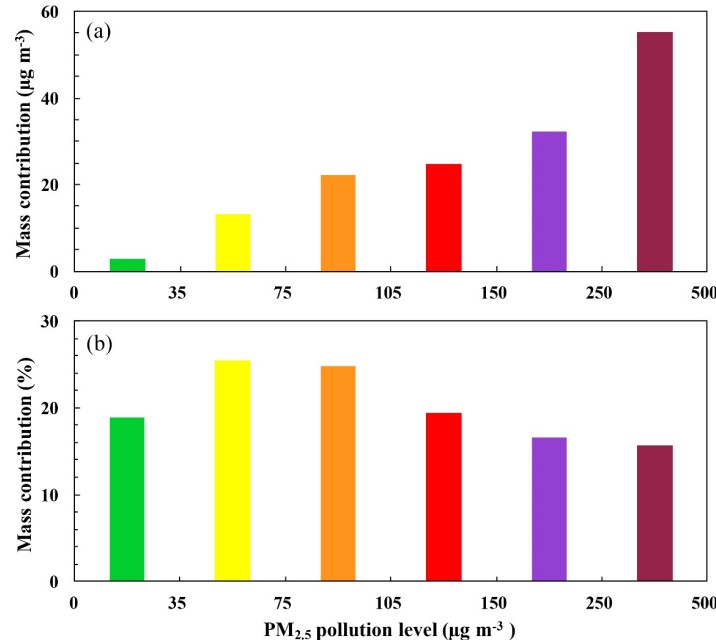

Figure 10 Average contributions of the RCC emission in Beijing to the local $PM_{2.5}$ mass
concentrations under different haze pollution levels from 9 to 25 January 2014. The green, yellow,
orange, red, purple, and dark red color bar in the plots represents excellent, good, slightly polluted,
moderately polluted, heavily polluted, and severely polluted levels of air quality, respectively.