# Peer review of "Contributions of residential coal combustion to the air quality in Beijing-Tianjin-Hebei (BTH), China: A case study"

_Atmospheric Chemistry and Physics, 2017_

## Referee Comment (RC1) · Anonymous Referee #3 · 27 Feb 2018

The paper by Li et al. quantifies the contributions of residential coal combustion to air quality in BTH in January 2014 to show the importance of controlling residential sources in reducing air pollution in the region. I have following major concerns needed to be addressed before the publication of the paper. 1. The paper lacks description of inputs to the model. For example, emission information is only mentioned in Table 1 with two papers cited. The two papers actually are two different inventories, which one is used. The emissions are annual/month averages. How did the authors do speciation, temporal allocation for different sources? The resolution of the emission is 0.25deg for MIX, how do you re-allocate them to 6km resolution for the model and what are the uncertainties regarding that?

[Figure]

2. The paper lacks validation of the model performance on meteorological conditions and air pollutants. The meteorological performance was not mentioned at all. For air pollutants, we can see from Figure 2, most sites have large differences between observation and predictions, mostly under-prediction. Figure 2 actually cannot show if the model performance is acceptable. Statistics should be based on those suggested by previous studies (see refs) are needed and comparison with other studies over China is important. Refs:

EPA, U.S.: Guidance on the Use of Models and Other Analyses in Attainment Demonstrations for the 8-hour Ozone NAAQS, EPA454/R-05-002, 2005.

J.W. Boylan, A.G. Russell. PM and light extinction model performance metrics, goals, and criteria for three-dimensional air quality models Atmos Environ, 40 (2006), pp. 4946-4959

Recommendations on statistics and benchmarks to assess photochemical model performance C Emery, Z Liu, AG Russell, MT Odman, G Yarwood, N Kumar Journal of the Air & Waste Management Association 67 (5), 582-598

3. The brute force method and the uncertainties are not mentioned. Due to the non-linear processes of atmospheric processes. Sensitivity methods such as brute force used in this study can tell the importance of the sources but have major flaws in quantifying the source contributions.

4. Not enough credits are given to previous source apportionment studies in China, especially the source-oriented models. See below examples:

https://doi.org/10.1016/j.envpol.2015.08.037

Source apportionment of PM2.5 for 25 Chinese provincial capitals and municipalities using a source-oriented Community Multiscale Air Quality model X Qiao, Q Ying, X Li, H Zhang, J Hu, Y Tang, X Chen Science of the Total Environment 612, 462-471

5. The writing can be improved, please go through more times. Limited examples are
given: a. WRF-Chem is used officially. b. Line 18, 9 to 25 January should a persistent episode c. "the" can be better used. For example, the Beijing-Tianjin-Hebei region, the Yangtze River Delta region. d. The reference method includes first name initial, not sure if ACP requires this now.
* * *

---

## Referee Comment (RC2) · Anonymous Referee #1 · 10 Mar 2018

WRF-Chem simulations with perturbed emissions are performed to quantify the contribution of residential coal combustion (RCC) to the particulate pollution in Beijing and surrounding region. The model shows good agreement with surface measurements on PM2.5 and speciated aerosol mass concentrations, which makes the following sensitivity simulations more reliable. The comparison of the RCC from Beijing versus the surrounding region provides a quantitative assessment of the efficiency of the residential coal replacement plans for the policy makers. The paper can be accepted by ACP after my following questions can be addressed.

1. The description of all the sensitivity experiments should be summarized in the Model and Methodology. More details should be provided what species in the emission inventories are turned off in each sensitivity run.

2. Total OA simulation is reported to be consistent with observations. Meanwhile, the authors mentioned the POA and SOA observations are available during the simulation time. It is interesting to know how OOA (representing SOA) is simulated in the model? In other words, is the primary:secondary ratio right for the aerosol sources in the model?

3. L279, was the electricity mainly from coal burning as well? L281, why the coal replacement plan in Beijing is controversial?

4. L333, "bring back the blue sky to Beijing" is a vague statement. What's the definition of "blue sky"? Better to use some criteria in term of PM level.

5. L340, the conclusion here is somewhat objective. 18% reduction can be considered "significant" as well. Please rephrase the sentence.

6. Is the atmospheric stability or air stagnancy changed by the coal emission as well? Light absorbing aerosols are thought to alter the ambient temperature profile locally [Wang et al., 2013; Zhang et al., 2015; Peng et al., 2016]. Your WRF-Chem simulations with aerosol-meteorology interactions should be able to answer such questions.  A related question is what is the TOA radiative forcing from RCC in your simulations?

7. Are the modeled CO spatiotemporal variations well correlated with total PM2.5 or a part of it like EC? Recently more observational studies use CO as an aerosol proxy to conduct aerosol related researches using the remote sensing technique.

8. Questions on the figures:
   • Figure 2, please thicken the circles in the plot, as they are hardly to see.
   • Figure 4, are they averaged over the 17 days from 9 to 25 January? If yes, I would expect to see a smoother diurnal variation in those plots. The spikes of OA and CCOA near 1800 look very sharp.
   • Figure 8, please specify what each color stands for in the figure caption.

9. Some grammar and English writing issues:

- L16, L316 assess contributions.
- L149, pollution simulations.
- L176, replicates.
- L184, reasonably yields.
- L272, from southwest to northeast. The usage of article is problematic in several places, please pay more attention.
- L292, still debatable.

---

## Referee Comment (RC3) · Anonymous Referee #2 · 16 Mar 2018

Review of "Contributions of residential 1 coal combustion to the air quality in Beijing-Tianjin-Hebei (BTH), China: A case study" by Xia et al.

General comments

The authors present a WRF-CHEM modeling study on quantifying the impacts of residential coal combustion (RCC) emissions on air quality in the BTH region. This study is done under the background that the BTH region has been plagued with persistent heavy haze pollution, coal combustion is a major pollutant emission source in this region, and there has been a debate on the roles of local emissions and regional transport in the haze pollution in Beijing. They conclude that although local RCC emissions make an important contribution to the haze pollution in Beijing, it is necessary to control the RCC emissions in the entire BTH and its surrounding areas in order to significantly reduce the haze levels in Beijing. The manuscript is well organized and presented, and the methodology is sound. It can be published with minor revisions.

Specific comments

1. According to the information in the Introduction (L81-91), there have been several model studies regarding the impacts of the emissions of coal combustion and/or RCC on air quality in the BTH. What are new in this study compared to these studies? Are the results of this study consistent with other studies, and if not, why?

2. When discussing the modeling discrepancy, the authors emphasize the bias from simulated meteorology. Emissions as another likely factor should also be addressed. The effect of the meteorology uncertainty should affect all pollutants, especially primary pollutants, not just PM2.5. As such, the authors should also examine if the earlier fall-off and underestimation occur to other pollutants (especially primary pollutants, such as CO and SO2); if it does, it provides additional evidence for the factor of meteorology; if not, other factors need to be taken into account.

3. L225, it would be helpful to provide the numbers by Huang et al (2014).

4. L268, I suggest to change the section title to something like "Contribution of local RCC emissions to air quality in Beijing" to differentiate section 3.3 and the case of "SEN-BTH".

5. L280-282, deliberate the "controversial issue".

L309, 25% contribution by local RCC does not warrant the RCC to be the MAIN cause.

L311 -312 and L336-337, you use the number of RCC contributing 15-20% to PM2.5 during moderate to severe pollution conditions to argue the importance of regional transport to the haze pollution in Beijing. There is a flaw in the argument, since there might be local anthropogenic emissions other than RCC that could make significant contributions too. To argue the importance of the regional transport, you better to contrast the results in the cases of BTH-SEN

and SEN-PEK (30% vs 18%, i.e., 12% from the RCC emission transport vs 18% from local RCC) to conclude.

Technique issues

1. The language needs to be polished.  Following are some examples

The use of "vice versa" in L206 and L225 is not correct. You mean "opposite"?
The use of "dispersion" in L186 and L199 is not appropriate either; you mean bias or disparity?
Delete "well" in L154, 173, 176, 184, 192 and 343;
In L21, 174, 177, L193, and L318, , change "compared with" or "compared to" to "when compared with" or "against".

2.  Delete the first name initials in L81-88

---

## Author Comment (AC1) · 18 Jun 2018

**Reply to Anonymous Referee #1**

We thank the reviewer for the careful reading of the manuscript and helpful comments. We have revised the manuscript following the suggestion, as described below.

WRF-Chem simulations with perturbed emissions are performed to quantify the contribution of residential coal combustion (RCC) to the particulate pollution in Beijing and surrounding region. The model shows good agreement with surface measurements on $PM_{2.5}$ and speciated aerosol mass concentrations, which makes the following sensitivity simulations more reliable. The comparison of the RCC from Beijing versus the surrounding region provides a quantitative assessment of the efficiency of the residential coal replacement plans for the policy makers. The paper can be accepted by ACP after my following questions can be addressed.

**1 Comment:** The description of all the sensitivity experiments should be summarized in the Model and Methodology. More details should be provided what species in the emission inventories are turned off in each sensitivity run.

**Response:** We have clarified the sensitivity simulations in Section 2.1 "*In the present study, we have conducted one reference simulation in which emissions from various anthropogenic and biogenic sources are considered (hereafter referred as to the REF case). The results from the REF case are compared with observations in BTH to validate the model performance. Additional two sensitivity simulations have also been performed, without the RCC emission in BTH and its surrounding areas and Beijing, respectively (hereafter referred as to the SEN-BTH case and SEN-PEK case). In the sensitivity simulation, the emission of $NO_x$, CO, VOCs, $SO_2$, black and organic carbon, primary sulfate, and unspecified particulate matters from the RCC is turned off. The difference between the reference and sensitivity simulation is used to evaluate contributions of RCC emissions to the air quality in BTH and Beijing.*".

**2 Comment:** Total OA simulation is reported to be consistent with observations. Meanwhile, the authors mentioned the POA and SOA observations are available during the simulation time. It is interesting to know how OOA (representing SOA) is simulated in the model? In other words, is the primary/secondary ratio right for the aerosol sources

in the model?

**Response:** We have clarified in Section 3.1.2 "*The modeled $PM_{2.5}$ mass concentration averaged during the simulation period in BTH and Beijing is 111.6 and 97.7 $\mu g$ $m^{-3}$, respectively. OA dominate the $PM_{2.5}$ in BTH, with a contribution of around 43.1%. Although the simulated $O_3$ concentration is low, the secondary aerosols, including SOA, sulfate, nitrate, and ammonium still make up about 40% of the $PM_{2.5}$ mass concentration, with contributions of 7.9%, 11.3%, 12.4%, and 9.6%, respectively. Elemental carbon and the unspecified aerosol species account for 7.5% and 16.2% of the $PM_{2.5}$ mass concentration, respectively. In Beijing, sulfate, nitrate, and ammonium constitutes 10.6%, 14.0%, and 9.1% of the $PM_{2.5}$ mass concentrations, respectively. OA are also the dominant constituent of the simulated $PM_{2.5}$ in Beijing, with a contribution of about 44.1%. The simulated ratio of the primary to secondary ratio OA in Beijing is 4.6, which is close to the observed ratio of 4.3.*"

**3 Comment:** L279, was the electricity mainly from coal burning as well? L281, why the coal replacement plan in Beijing is controversial?

**Response:** We have clarified in Section 3.3 "*It is worthy to note that the electricity is principally from the coal burning in China, and the main air pollutants emitted from coal-burning power plants are $NO_x$ and $SO_2$. However, the major pollutants emitted by the residential coal combustion include organic carbon, $SO_2$ and $NO_x$. Considering the dominant role of OA in the $PM_{2.5}$ in Beijing, the coal replacement in residential living is more effective in power plants.*".

**4 Comment:** L333, "bring back the blue sky to Beijing" is a vague statement. What's the definition of "blue sky"? Better to use some criteria in term of PM level.

**Response:** We have modified the expression in Section 4 "*Hence, the coal replacement plan in Beijing is beneficial to the local air quality, but is not as anticipated to substantially improve the air quality.*".

**5 Comment:** L340, the conclusion here is somewhat objective. 18% reduction can be considered "significant" as well. Please rephrase the sentence.

**Response:** We have rephrased the sentence in Section 4 "*Our results indicate that if the residential coal replacement is only implemented in Beijing, Beijing's air quality will be improved considerably, but not substantially, considering the impact of trans-boundary transport.*".

**6 Comment:** Is the atmospheric stability or air stagnancy changed by the coal emission as well? Light absorbing aerosols are thought to alter the ambient temperature profile locally [Wang et al., 2013; Zhang et al., 2015; Peng et al., 2016]. Your WRF-Chem simulations with aerosol-meteorology interactions should be able to answer such questions. A related question is what is the TOA radiative forcing from RCC in your simulations?

**Response:** We have clarified in Section 3.2 "*It is worth noting that light absorbing aerosols are thought to alter the ambient temperature profile locally (Wang et al., 2013; Zhang et al., 2015; Peng et al., 2016). The sensitivity results indicate that if the RCC emission in BTH and its surrounding areas is excluded, the surface temperature in BTH is decreased by about 0.23℃ on average during the study period, about half of which is contributed by light absorbing aerosols.*".

**7 Comment:** Are the modeled CO spatiotemporal variations well correlated with total $PM_{2.5}$ or a part of it like EC? Recently more observational studies use CO as an aerosol proxy to conduct aerosol related researches using the remote sensing technique.

**Response:** The modeled CO spatiotemporal variations is well correlated with the $PM_{2.5}$ and we have further correlated CO with $PM_{2.5}$ mass concentrations in Section 3.1.1 and Figure 5: "*Recently, observational studies have used CO as an aerosol proxy to investigate atmospheric aerosols based on the remote sensing technique. Figure 5 shows the scatter plots of observed and simulated $PM_{2.5}$ with CO mass concentrations averaged over all ambient monitoring sites in BTH during the simulation period. The observed and simulated CO mass concentrations are well correlated with those of $PM_{2.5}$, with the $R^2$ exceeding 0.81.*".

**8 Comment:** Questions on the figures:

• Figure 2, please thicken the circles in the plot, as they are hardly to see.

**Response:** We have modified Figure 2 (now Figure 3) as suggested.

• Figure 4, are they averaged over the 17 days from 9 to 25 January? If yes, I would expect to see a smoother diurnal variation in those plots. The spikes of OA and CCOA near 1800 look very sharp.

**Response:** Figure 4 (Now Figure 6) shows the diurnal variations of the observed and simulated aerosol species from 9 to 25 January. The spikes of OA and CCOA are caused by the irregular residential emissions near the observation site.

• Figure 8, please specify what each color stands for in the figure caption.

**Response:** We have clarified in the caption of Figure 8 (Now Figure 10): "*The green, yellow, orange, red, purple, and dark red represents excellent, good, slightly polluted, moderately polluted, heavily polluted, and severely polluted levels of air quality, respectively.*".

**9 Comment:** Some grammar and English writing issues:
• L16, L316 assess contributions.
• L149, pollution simulations.
• L176, replicates.
• L184, reasonably yields.
• L272, from southwest to northeast. The usage of article is problematic in several places, please pay more attention.
• L292, still debatable.

**Response:** We have revised the manuscript as suggested.

---

## Author Comment (AC2) · 18 Jun 2018

**Reply to Anonymous Referee #2**

We thank the reviewer for the careful reading of the manuscript and helpful comments. We have revised the manuscript following the suggestion, as described below.

The authors present a WRF-CHEM modeling study on quantifying the impacts of residential coal combustion (RCC) emissions on air quality in the BTH region. This study is done under the background that the BTH region has been plagued with persistent heavy haze pollution, coal combustion is a major pollutant emission source in this region, and there has been a debate on the roles of local emissions and regional transport in the haze pollution in Beijing. They conclude that although local RCC emissions make an important contribution to the haze pollution in Beijing, it is necessary to control the RCC emissions in the entire BTH and its surrounding areas in order to significantly reduce the haze levels in Beijing. The manuscript is well organized and presented, and the methodology is sound. It can be published with minor revisions.

**1 Comment:** According to the information in the Introduction (L81-91), there have been several model studies regarding the impacts of the emissions of coal combustion and/or RCC on air quality in the BTH. What are new in this study compared to these studies? Are the results of this study consistent with other studies, and if not, why?

**Response:** We have clarified in Section 1: "*Control strategies have also been implemented to reduce residential emissions, but evaluation means constrained by observations are still lacking.*" and "*Until now, there have been few studies focusing specially on the impacts of RCC emissions on the air quality in BTH.*". The results of this study are not consistent with those of other studies which generally evaluate the contribution of residential living emissions to the air quality in China.

**2 Comment:** When discussing the modeling discrepancy, the authors emphasize the bias from simulated meteorology. Emissions as another likely factor should also be addressed. The effect of the meteorology uncertainty should affect all pollutants, especially primary pollutants, not just $PM_{2.5}$. As such, the authors should also examine if the earlier fall-off and underestimation occur to other pollutants (especially primary

pollutants, such as CO and SO$_2$); if it does, it provides additional evidence for the factor of meteorology; if not, other factors need to be taken into account.

**Response:** We have clarified the impact of uncertainties in emission inventory and meteorology to the modeling biases in Section 3.1.1 "*The early occurrence of intensified winds in simulations also cause rapid falloff of SO$_2$ and CO mass concentrations during the haze dissipation stage. Besides uncertainties in meteorological field simulations, uncertainties in emission inventory are also responsible for the model biases of air pollutants. Since implementation of the APPCAP, strict emission control measures have been made to improve the air quality in BTH, and the spatiotemporal variations of anthropogenic emissions in BTH have changed considerably (Li et al., 2017), which is not reflected in the emission inventory used in the present study.*".

**3 Comment:** L225, it would be helpful to provide the numbers by Huang et al (2014).

**Response:** We have clarified in Section 3.1.2 "*The simulated chemical composition in Beijing is generally comparable to the observation in January 2013 by Huang et al. (2014), showing that OA constitutes a major fraction (40.7%) of the total PM$_{2.5}$, followed by sulfate (16.0%), nitrate (12.0%), and ammonium (9.8%). It is worth noting that the simulated sulfate contribution to PM$_{2.5}$ mass concentrations in Beijing is lower than the observation in Huang et al. (2014), and vice versa for the nitrate aerosol. Implementation of the APPCAP since 2013 September has considerably decreased SO$_2$ emissions in BTH, lowering the sulfate formation. Additionally, the decrease of the sulfate aerosol reduces its competition with ammonia in the atmosphere, facilitating the nitrate formation.*".

**4 Comment:** L268, I suggest to change the section title to something like "Contribution of local RCC emissions to air quality in Beijing" to differentiate section 3.3 and the case of "SEN-BTH".

**Response:** We have changed the section title of 3.3 to "*Contributions of local RCC emission to the air quality in Beijing*".

**5 Comment:**

• L280-282, deliberate the "controversial issue".

**Response:** We have Clarified in Section 3.3 "*It is worthy to note that the electricity is principally from the coal burning in China, and the main air pollutants emitted from coal-burning power plants are $NO_x$ and $SO_2$. However, the major pollutants emitted by residential coal combustion include organic carbon, $SO_2$ and $NO_x$. Considering the dominant role of OA in the $PM_{2.5}$ in Beijing, the coal replacement in residential living is more effective in power plants.*".

• L309, 25% contribution by local RCC does not warrant the RCC to be the MAIN cause.

**Response:** We have clarified in Section 3.3 "*Apparently, the mitigation effect is the best under good and lightly polluted conditions in terms of $PM_{2.5}$ level, and the $PM_{2.5}$ mass concentration decreases by around 25% when the RCC emission in Beijing is not considered, indicating that the local RCC emission does not constitute the main $PM_{2.5}$ pollution source in Beijing.*".

• L311 -312 and L336-337, you use the number of RCC contributing 15-20% to $PM_{2.5}$ during moderate to severe pollution conditions to argue the importance of regional transport to the haze pollution in Beijing. There is a flaw in the argument, since there might be local anthropogenic emissions other than RCC that could make significant contributions too. To argue the importance of the regional transport, you better to contrast the results in the cases of BTH-SEN and SEN-PEK (30% vs 18%, i.e., 12% from the RCC emission transport vs 18% from local RCC) to conclude.

**Response:** We have clarified in Section 3.3 "*Sensitivity studies show that when only the RCC emission in Beijing is excluded in simulations, the $PM_{2.5}$ level is decreased by 18%, much less than about 30% decrease caused by the exclusion of the RCC emission in BTH and its surrounding areas, showing the important contribution of trans-boundary transport to the air quality in Beijing.*".

**6 Comment:** The language needs to be polished. Following are some examples:

• The use of "vice versa" in L206 and L225 is not correct. You mean "opposite"?

**Response:** We have changed "vice versa" to opposite as suggested.

• The use of "dispersion" in L186 and L199 is not appropriate either; you mean bias or disparity? Delete "well" in L154, 173, 176, 184, 192 and 343;

**Response:** We have changed "dispersion" to "deviation" in Section 3.1.1. We have deleted "well" as suggested.

• In L21, 174, 177, L193, and L318, change "compared with" or "compared to" to "when compared with" or "against".

**Response:** We have changed "compared with" to "compared to" or "against" as suggested.

**7 Comment:** Delete the first name initials in L81-88

**Response:** We have deleted the first name initials as suggested.

---

## Author Comment (AC3) · 18 Jun 2018

**Reply to Anonymous Referee #3**

We thank the reviewer for the careful reading of the manuscript and helpful comments. We have revised the manuscript following the suggestion, as described below.

The paper by Li et al. quantifies the contributions of residential coal combustion to air quality in BTH in January 2014 to show the importance of controlling residential sources in reducing air pollution in the region. I have following major concerns needed to be addressed before the publication of the paper.

**1 Comment:** The paper lacks description of inputs to the model. For example, emission information is only mentioned in Table 1 with two papers cited. The two papers actually are two different inventories, which one is used. The emissions are annual/month averages. How did the authors do speciation, temporal allocation for different sources? The resolution of the emission is 0.25deg for MIX, how do you re-allocate them to 6km resolution for the model and what are the uncertainties regarding that?

**Response:** We have clarified in Section 2.1 "*The WRF-Chem model adopts one grid with a horizontal resolution of 6 km centered at 39°N, 117°E, and 35 sigma vertical levels with a stretched vertical grid with spacing ranging from 30 m near the surface, to 500 m at 2.5 km and 1 km above 14 km, and the grid cells used for the domain are 150 × 150. The physical parameterizations employed in the simulation include the microphysics scheme of Hong and Lim (2006), the unified Noah Land-surface model (Chen and Dudhia, 2001), the Goddard longwave scheme (Chou and Suarez, 2001), and the Goddard shortwave scheme (Chou and Suarez, 1999). The National Centers for Environmental Prediction (NCEP) 1°×1° reanalysis data are used for the meteorological initial and boundary conditions, and the meteorological simulations are not nudged in the study. The chemical initial and boundary conditions are interpolated from the 6 h output of MOZART (Horowitz et al., 2003). The spin-up time of the WRF-Chem model is 28 h. The monthly average anthropogenic emissions with a 6 km horizontal resolution in the North China Plain are developed by Zhang et al. (2009) with the base year of 2013, including contributions from agriculture, industry, power generation, residential, and transportation sources, and the volatile organic compounds (VOCs) speciation based on the SAPRC chemical mechanism. The*

*temporal allocation for different sources follows those in Zhang et al. (2009). The biogenic emissions are calculated online using the MEGAN (Model of Emissions of Gases and Aerosol from Nature) model developed by Guenther et al. (2006).".*

**2 Comment:** The paper lacks validation of the model performance on meteorological conditions and air pollutants. The meteorological performance was not mentioned at all. For air pollutants, we can see from Figure 2, most sites have large differences between observation and predictions, mostly under-prediction. Figure 2 actually cannot show if the model performance is acceptable. Statistics should be based on those suggested by previous studies (see refs) are needed and comparison with other studies over China is important. Refs:

EPA, U.S.: Guidance on the Use of Models and Other Analyses in Attainment Demonstrations for the 8-hour Ozone NAAQS, EPA454/R-05-002, 2005.

J.W. Boylan, A.G. Russell. PM and light extinction model performance metrics, goals, and criteria for three-dimensional air quality models Atmos Environ, 40 (2006), pp. 4946-4959.

Recommendations on statistics and benchmarks to assess photochemical model performance C Emery, Z Liu, AG Russell, MT Odman, G Yarwood, N Kumar Journal of the Air & Waste Management Association 67 (5), 582-598.

**Response:** We have made further validations of meteorological fields and air pollutants in Figure 2 and Table 2 as suggested and clarified in Section 3.1.1 "*Considering the key role of meteorological fields in determining the formation transformation, diffusion, transport, and removal of the air pollutants, Figure 2 presents the diurnal profiles of the observed and simulated temperature, relative humidity (RH), wind speed and direction at meteorological sites in Beijing, Tianjin, and Shijiazhuang during the simulation period. The WRF-Chem model reasonably well predicts the diurnal variations of the temperature in the three cities against observations, with IOAs of around 0.80. The model also well yields the temporal variation of the RH in Beijing when compared with observations, but it tends to underestimate the RH in Tianjin and Shijiazhuang with IOAs less than 0.70, and generally fails to capture the high RH exceeding 80%. The temporal variations of the wind speed and direction in BTH are also reasonably reproduced, but the model biases are still rather large.*".

"*Table 2 presents the further validation of WRF-Chem model simulations of air*

*pollutants based on statistics methods suggested by previous studies (US EPA, 2005; Boylan and Russell, 2006; Emery et al., 2017). Compared to the suggested model performance criteria of air pollutants, the WRF-Chem model performs well in simulating the air pollutants and aerosol species in this study. The FB, FE, NMB, and NME of $PM_{2.5}$ and $O_3$ are generally within the benchmarks, with the correlation coefficients approaching 0.90, showing good consistency between the simulations and observations. As for the aerosol species, except for sulfate, the differences between the observed and simulated organic aerosol, nitrate, and ammonium are all less than the reference criteria. The FB and FE of sulfate are reasonable, but the NMB of 37.6% and NME of 67.8% are slightly higher than the suggested criteria.".*

**3 Comment:** The brute force method and the uncertainties are not mentioned. Due to the non-linear processes of atmospheric processes. Sensitivity methods such as brute force used in this study can tell the importance of the sources but have major flaws in quantifying the source contributions.

**Response:** We have clarified the brute force method in Section 2.1 "*The brute force method is used to quantify the contribution of the RCC emission in BTH and its surrounding areas to the air quality (Dunker et al., 1996). It is worth noting that, although the method can evaluate the importance of the certain emission source, it still has flaws in quantifying the source contribution, considering the complicated non-linear processes in the atmosphere (Zhang and Ying, 2011).*".

**4 Comment:** Not enough credits are given to previous source apportionment studies in China, especially the source-oriented models. See below examples:
https://doi.org/10.1016/j.envpol.2015.08.037.
Source apportionment of $PM_{2.5}$ for 25 Chinese provincial capitals and municipalities using a source-oriented Community Multiscale Air Quality model. X Qiao, Q Ying, X Li, H Zhang, J Hu, Y Tang, X Chen. Science of the Total Environment, 612, 462-471.

**Response:** We have clarified in Section 1 "*Using the source-oriented CMAQ model, Qiao et al. (2017) have conducted simulations to evaluate source apportionment of $PM_{2.5}$ in 25 Chinese provincial capitals and municipalities and concluded that industrial and residential sources are predicted to be the largest contributor to $PM_{2.5}$*

*for all the city groups, with annual fractional contributions of 25.0%-38.6% and 9.6%-27%, respectively.".*

**5 Comment:** The writing can be improved, please go through more times. Limited examples are given:

a. WRF-Chem is used officially.

**Response:** We have changed "WRF-CHEM" to "WRF-Chem" as suggested.

b. Line 18, 9 to 25 January should a persistent episode.

**Response:** We have changed it to "a persistent episode".

c. "the" can be better used. For example, the Beijing-Tianjin-Hebei region, the Yangtze River Delta region.

**Response:** We have revised the manuscript as suggested.

d. The reference method includes first name initial, not sure if ACP requires this now.

**Response:** We have revised the manuscript as suggested.